

# The importance of detection thresholds for the quantification of source and timing of high-latitude dust emission using remote sensing

Rosemary Huck[1], Robert G. Bryant[2], James King[3]

[1]School of Geography and the Environment, Oxford University Centre for the Environment, University of Oxford, Oxford OX1 3QY, UK
[2]Department of Geography, University of Sheffield, Sheffield S10 2TN, UK
[3]Laboratoire d'Érosion Éolienne, Département de Géographie, Université de Montréal, Montréal, H2V 0B3, Canada

*Correspondence to*: Rosie Huck - rosemary.huck@ouce.ox.ac.uk

**Abstract.**

The observation and quantification of mineral dust fluxes from high-latitude sources remains difficult due to a known paucity of year-round in situ observations and known limitations of satellite remote sensing data (e.g., cloud cover and dust detection). Here we explore the chronology of dust emissions at a known and instrumented high latitude dust source: Lhù'ààn Mân (Kluane Lake) in Yukon, Canada. At this location we combine ground instrumentation, space-based remote sensing platforms, ground-based AERONET data, and oblique camera images to (i) investigate the daily to annual chronology of dust emissions recorded by these instrumental and remote sensing methods (at timescales ranging from minutes to years), and (ii) use data intercomparisons to comment on the principal factors that control the detection of dust in each case. Dust emissions were observed using oblique time-lapse (RC) cameras installed at Lhù'ààn Mân for up to 23 hours a day. These were used as a baseline for analysis of aerosol retrievals from in situ metrological data, AERONET, and co-incident MODIS MAIAC.

Use of high-cadence remote camera (RC) data collected during dust events allowed us to optimise the use of combination of date quality (DQ) 1 (aerosol optical depth - AOD) and DQ2 (single scattering albedo and Angstöm exponent) to best represent AOD dust retrievals from AERONET. Nevertheless, when compared with time series of RC data, optimised AERONET data only manage an overall 26 % detection rate for events (sub day) but 100% detection rate for dust event days (DED) when dust was within the field of view. Here, in this instance, RC and remote sensing data were able to suggest that the low event detection rate was attributed to fundamental variations in dust advection trajectory, dust plume height, and inherent restrictions in sun angle at high latitudes. Working with a time series of optimised AOD data (covering 2018/2019), we were able to investigate the gross impacts of DQ choice on DED detection at the month/year scale. Relative to ground observations, AERONET's DQ2.0 cloud screening algorithm may remove as much as 97 % of known dust events (3% detection). Finally, when undertaking an AOD comparison for DED and non-DED retrievals, we find that cloud screening of MODIS/AERONET lead to a combined low sample of co-incident dust events, and weak correlations between retrievals. Our results quantify and explain the extent of under-representation of dust in both ground and space remote sensing method; a factor which impacts on the effective calibration and validation of global climate and dust models.

Keywords: High-latitude dust; aerosols; AERONET; Planet Labs; MAIAC; wind erosion



## 1 Introduction

Mineral aerosols (MA or dust) are a large source of uncertainty in assessments and predictions of climate change (IPCC, 2013) and the importance dust sources in the high-latitude ($\geq$ 50 °N and $\geq$ 40 °S, including Arctic as a subregion $\geq$ 60 °N) in land-ocean-atmosphere systems has recently been recognised (IPCC, 2019; Meinander et al., 2022; Schmale et al., 2021; Tobo et al., 2019; Groot Zwaaftink et al., 2016). Plumes of dust are known as dust events and when dust is suspended in the atmosphere it can change the scattering and absorption of incoming solar radiation, reducing the amount of incoming solar radiation (Arnalds et al., 2016; Haywood and Boucher, 2000; Kylling et al., 2018; Yoshioka et al., 2007) and affect local meteorology through altering cloud properties by becoming condensation or ice nuclei triggering cloud formation (Murray et al., 2021; Tobo et al., 2019; Xi et al., 2022). Deposited dust are important sources of nutrients in both aquatic (Crusius et al., 2017; Schroth et al., 2017) and terrestrial (Moroni et al., 2018) ecosystems.

The impact that dust suspended in the atmosphere has on the climate system varies with dust characteristics such as particle shape, size, and composition (Kok, 2011; Bryant, 2013; Bullard et al., 2016), all of which are highly influenced by their origin. Some high-latitude mineral aerosols (HLMA) can be distinguished from low-latitude mineral aerosols through different physical, optical, and chemical properties, owing to their glaciogenic beginnings (Bachelder et al., 2020; Baldo et al., 2020; Crocchianti et al., 2021; Crusius, 2021; Meinander et al., 2022). High-latitude dust also have an important role influencing ice and snow albedo, with dust depositing onto glaciers and ice sheets forming cryoconite and subsequently influencing glacier hydrology, mass balance, and inducing cryospheric melt (Cooke et al., 2016; IPCC, 2019; Krinner et al., 2006; Réveillet et al., 2022; Wientjes et al., 2011). With increased deglaciation deposition of dust onto ice has been increasing, indicating a greater influence of high-latitude dust on the future climate system (Amino et al., 2021; Shi et al., 2022).

Although an estimated 5% of global dust emissions originate from high-latitude sources (Bullard et al., 2016; Groot Zwaaftink et al., 2016), the periods when emissions occur are highly seasonal (van Soest et al., 2022). The supply of glacially derived sediment is dependent upon glacier dynamics and related hydrology (Bullard et al., 2011). Spatially explicit sources are mostly de-glaciated valleys that have an abundance of fine, glacially derived, sediment known as 'glacial flour' providing near-perfect conditions for entrainment, linked to a particle size mean of ~100 µm (e.g., Mockford et al., 2018). But emissions are also temporarily limited to when the snow has melted, the ground has thawed, and the rivers are low, for when sediment can be made available for transport by wind (e.g., Crusius et al., 2011). The inherent challenges of operating in these polar regions (including inaccessibility and expense) and capturing events have led to, aside from a few well-monitored sites, high-latitude emission locations being poorly instrumented, and emission characteristics largely being derived from remotely sensed platforms (Ranjbar et al. 2020; van Soest et al., 2022).

Ultimately, global dust models can struggle to predict the amount of dust emitted from high-latitude sources (e.g., Schmale et al., 2021). Crusius (2021) suggests that this may be due to challenges of matching model spatial resolution to the scale at which complex thermally driven, high-speed winds which are topographically funnelled down narrow proglacial valleys and operate to drive dust emissions. At the same time, detection, and quantification of high-latitude dust events and dust sources is known to be challenging (Bullard et al., 2016). Dust





events are often detected through increases in aerosol optical depth (AOD), whether this be through ground (e.g., AERONET) or space-based (e.g., MODIS products) methods. Both approaches are impacted by cloud and the discrete nature of events in space and time (Bryant and Baddock, 2022). Nevertheless, improved detection,
understanding, and quantification of high-latitude dust events remains integral to the development and accuracy of global dust and climate models (Arnalds et al., 2016; Bullard et al., 2016).

Dust particles suspended in the atmosphere often have a short lifetime and are heterogeneous in nature (Schepanski, 2018). They are difficult to detect and characterise, especially using space-based techniques (Ciren
and Kondragunda, 2014; Kahn et al., 2015; Luo et al., 2015). Consequently, ground-based detection techniques have become a useful aid to verify and improve the accuracy of satellite retrievals, as well as a foundation for independent atmospheric investigation (Estavan et al., 2019; Formenti et al., 2011). The Aerosol Robotic Network (AERONET) is a ground-based collaborative network of automated sun-sky scanning spectral radiometers that determine the aerosol optical and microphysical properties by direct sun measurements (Holben, 1998).
AERONET is often used in comparison to space-based aerosol detection techniques, as it is used as a standard means of data calibration and validation (Martins et al., 2017; Mhawish et al., 2019). The spectral aerosol optical depth determined from these data are also used to derive an Angström exponent (α) which can in turn allow estimation of aerosol size distributions (O'Neill et al., 2003). These data have been used to differentiate between coarse-mode particles (dust) and fine-mode particles (such as smoke and anthropogenic pollutants; Eck et al.,
1999; Ciren and Kondragunta, 2014).

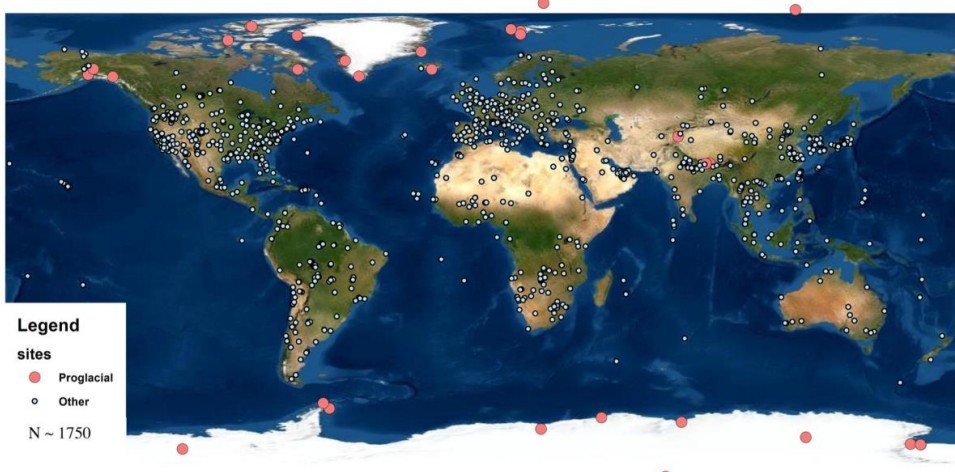

**Figure 1. Location of all AERONET stations with stations in the high-latitudes or proglacial areas highlighted in yellow. Cryospheric stations account for 31 out of ~1075 global AERONET stations (data from the AREONET website: https://aeronet.gsfc.nasa.gov) .**


Understandably, the small number of high-latitude AERONET stations relative to lower-latitude stations (3%; see figure 1) has made studies of the radiative properties of HLMA for both calibration and validation of models and optimisation of detection algorithms difficult. The most readily available resource for the processes of detection and quantification of HLMA events and advection are a suite of Low Earth Orbit (LEO) sensors such as Moderate



Resolution Imaging Spectroradiometer (MODIS) onboard the Aqua and Terra NASA satellites (e.g., Bryant, 2013; Kandakji et al., 2020; McGrath et al., 2010; Mhawish et al., 2019). MODIS has been used successfully to monitor single HLMA events at Ellesmere Island, Nunavut, Canada, (Ranjbar et al., 2020) but consistent monitoring of high latitude dust sources using LEO sensors is poses specific challenges. Importantly, there are some identifiable spatial and temporal bias inherent in remote sensing of high latitude dust using LEO sensors,

which includes an understanding of the impact of clouds on dust event and dust source detection (e.g., Gasso and Stein, 2007) and the fact that most polar orbiting LEO platforms/sensors have overpass times that may observe dust in transport (i.e. dust advection) rather than active emission near to source; or not at all. It is clear, for example, that remote sensing observations of HLMA are significantly under-estimated as a result (Prospero et al., 2012; Bullard et al., 2016). Where possible satellite observations are often augmented with ground-based meteorological

and dust flux data (e.g., Arnalds, 2010; Schroth et al., 2017; Thorsteinsson et al., 2011) and occasionally verified through use of ground-based AERONET (Baddock et al., 2009; Harley et al., 2017). As a result, our understanding of the spectral signatures of HLMA in AERONET data remains important, as the few well monitored research areas that have AERONET stations in the high latitudes are necessarily used as analogues for other HLMA emission locations (e.g., Shi et al., 2022).


Here we seek to improve our understanding of high-latitude dust event detection in space and time. Our research focuses on dust event detection at a known persistent dust source in the proglacial area of Lhù'ààn Mân (Kluane Lake) in the Yukon, Canada, and involves synergistic analysis of dust aerosol optical properties via both space and ground based remote sensing platforms to characterise dust event occurrence and locate dust sources. As part

of this endeavour, we seek to address the following research questions,

1. What is the annual chronology of high-latitude mineral aerosol (HLMA) emissions at Lhù'ààn Mân (Kluane Lake) as observed via different remote sensing retrieval approaches (LEO, and ground based AERONET)?

2. How do remote observations of dust compare to ground (Met, RC) and AERONET observations at the

high-latitude dust event scale?

## 2 Site Information

The proglacial area surrounding Lhù'ààn Mân (61° 1' 5" N, 138° 29' 40"W), Yukon, provides an exceptional opportunity for quantifying and detecting dust events where events are not supply limited. The site is uniquely heavily instrumented with a suite of ground-based environmental monitoring and remote sensing platforms. Á'áy

Chù (Slims River) is a proglacial river from the Kaskawulsh Glacier, in the Yukon, Canada. The river delta is ~25 km from the glacier terminus and until recently, flowed into Lhù'ààn Mân. The soils in the 8 km wide valley consist mostly of glacial flour and are subject to strong, topographically funnelled and thermally driven winds (Bachelder et al., 2020).





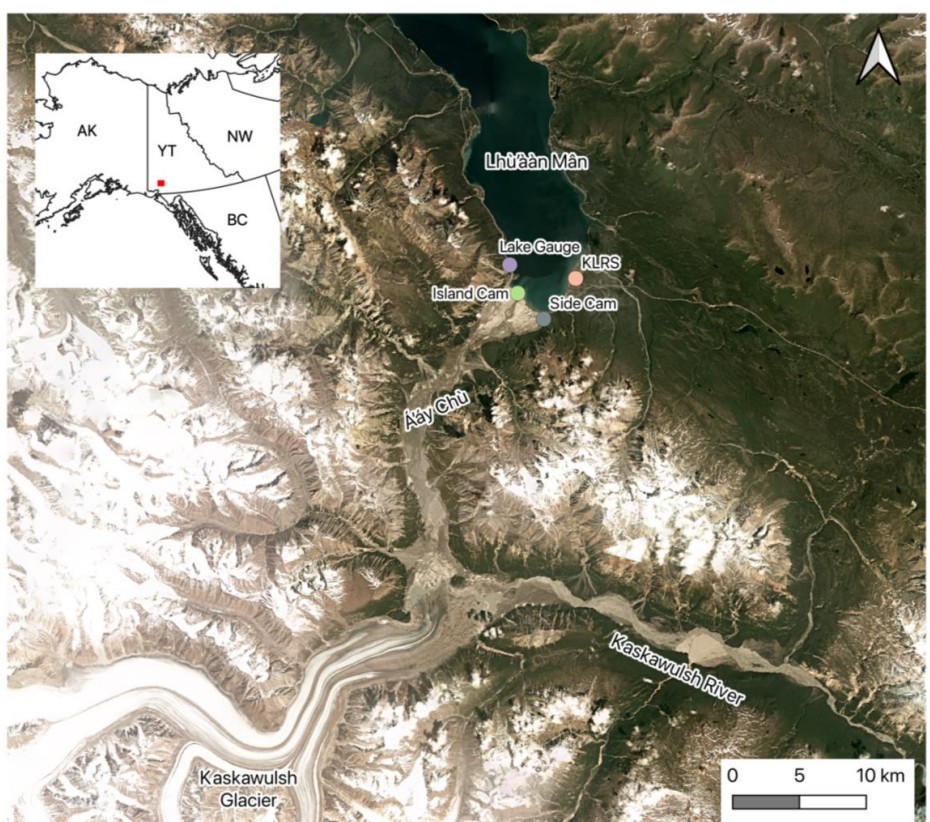


**Figure 2. Lhù'ààn Mân study site and setting with notable measuring stations and features highlighted. Kluane Lake Research Station (KLRS) contains both the AERONET and meteorological stations. Base imagery is from Planet imagery in June 2018.**

**2.1 Climatology**

The region is in the shadow of some of the highest peaks in North America (e.g., Mt Logan, 5959 m above sea level). Subsequently the Lhù'ààn Mân region is in a prominent orographic rain shadow by the St. Elias Mountain, and has a dry, cold continental climate with an annual precipitation of 300–800 mm (Williamson et al., 2014). As with many of the large lakes in the Yukon, historical scientific data around Lhù'ààn Mân is limited (McKnight et

al., 2021). A weather station was established at Burwash Landing at the northern end of the lake in 1967, however the station is ~45km from the research site and strong meteorological gradients towards the alpine glaciers produce a highly variable climate in the region. The climate of southwestern Yukon was warmer than it is today in the late Holocene and Lhù'ààn Mân discharged south towards the Alsek River. Climate statistics collected at the Lhù'ààn Mân research site in 2018 show an average temperature of 8.7 ±6.4 °C during May to October with a 0 °C crossing

occurring in late April.



### 2.2 Glacial Retreat and Riverbed Exposure

The Kaskawulsh glacier (Figure 2) has been in retreat since the nineteenth century, with retreat increasing in the late twentieth and early twenty-first centuries, with the glacier retreating 655m from 1956-2007 (Foy et al., 2011; Shugar et al., 2017). The retreat led to the Á'áy Chù being the first recorded river to be impacted by climate change as the meltwaters were redirected into the eastward flowing Kaskawulsh river, dramatically reducing the Á'áy Chù's discharge and leaving more riverbed exposed for aeolian sediment deflation (Shugar et al., 2017; Bachelder et al., 2020). The re-routing of the river has resulted in a significant increase in the frequency of dust events at the river delta.

### 2.3 Prior Dust Observations

The delta of the Á'áy Chù is known to be a persistent dust source with events observed daily during the spring and summer during previous field research (Nickling, 1978; Nickling and Brazel, 1985; Bachelder et al., 2020). As a result, the site has been carefully and highly instrumented to understand dust emissions. In 2018 during a month-long field trip in May, dust in motion was observed almost daily and emitted dust exceeded local air quality safety thresholds (Bachelder et al., 2020). Dust emitted at the site comprises primarily non-spherical aluminosilicate clay mineral aggregates, as well as other pure minerals (Bachelder et al., 2020). As over half the sediment entrained at the site are composed of clay, the suggested predominant mechanism of dust emission is through the rupturing of clay coatings and release of resident fine particles (Bachelder et al., 2020). This site is highly dynamic, where glacial, fluvial, and aeolian processes meet and interplay and are stressed by climate warming and typical of dust emissions in the Yukon. However, the relative lack of observations of high-latitude dust in North America compared to other high latitude sources (e.g., Iceland and Greenland) results in the specific need for enhanced remote sensing data monitoring approaches and forecast models to fully understand the contribution of North American high latitude dust in the global dust cycle.

## 3 Data and Methods

### 3.1 Meteorological Measurements

Half hourly maximum wind speed, average wind direction, rainfall total, atmospheric pressure, relative humidity, snow depth, and average temperature are recorded at Kluane Lake Research Station (KLRS). Lake depth is taken from the Environment Canada recording station Kluane Lake near Burwash Landing (9CA001). Following the river diversion in 2016, Lhù'ààn Mân has been receding in height due to the decreased discharge of the Á'áy Chù with maximum lake depth dropping from 4.51m in 2013 to 2.20 m in 2016 at the Burwash Landing recording site. In this region, there are quite complex wind regimes which are influenced by the topography, the ice caps on the mountains, and the presence of Lhù'ààn Mân. The meteorological station at KLRS is located on the southern end of the lake ~ 7 km east of the river-lake confluence. This location means that the station is influenced by stronger synoptic winds travelling down the valley and across the lake, however thermal drainage winds channel down the steep sided valleys and influence wind direction at the study site. This is evidenced in meteorological stations further up and down the valley which are much less directionally variable than KLRS, with dominant wind directions of North-Northeast (see supplementary information). The average wind speed over the 2018 dust season at KLRS was 3.60 m s$^{-1}$ varying from 0.00 to 15.01 m s$^{-1}$.



### 3.2 AERONET Observations

The AERONET (Aerosol Robotic Network; Holben, 1998) is a network of ground-based sun photometers that measure the rate of solar ray extinction in the atmospheric column above the photometer to determine AOD alongside other atmospheric properties. Data from the network of sun photometers is uploaded in near-real time (within two hours of being recorded) and is freely available for download from NASA's AERONET website (https://aeronet.gsfc.nasa.gov) processed by the Version 3 automated control algorithm (Giles et al., 2019). Both AOD data and α (Eck et al., 1999) AERONET data are often available for immediate or near-real time download.

The Kluane Lake AERONET station (see Figure 2 for location) records data from early May to late October each year. AERONET returns at 1.0 data quality (DQ) range for between 252 returns per day (maximum observed) to 2 returns per day (minimum observed). At the study site, dust events were observed and recorded throughout the May-October period, along with the presence of other aerosols (e.g., smoke and marine). AERONET AOD data are computed at three DQ levels: Level 1.0 (unscreened), level 1.5 (cloud-screened and quality assured), and Level 2.0 (quality assured) with the Version 3 automated control algorithm. For this study, all DQ level AOD returns at 1020 nm and 500 nm were used and evaluated, and all AERONET data were derived using the Version 3 retrieval algorithm. The absolute error in AERONET AOD retrievals at all wavelengths used in this study is assumed to be 0.01-0.02 (Holben et al., 1998; AERONET website).

The primary AERONET station used in this study is located at the Kluane Lake Research Station (KLRS) at Silver City Airport (61° 1'38.14" N 138°24'38.58" W). This is at the southern end of the lake ~7 km east of the river-lake confluence (see Figure 2). As the sun photometer undertakes almucantar retrievals during daylight hours, dust entrained from the delta that interacts with the scanning field-of-view is recorded. As noted, other aerosol pollutants also contribute to AOD at this site, so to discern the presence of dust, a combination of AERONET observation at 1020 nm with an AOD greater than 0.3 and an α of less than 0.6 between 440 nm and 870 nm was used (Dubovik et al., 2002). In this case, we use detection at 1020 nm as the regression curves of optical parameters are more robust than those of other wavelengths for suspended dust (Dubovik et al., 2002). Days in which these criteria were met in the AERONET 1.0 DQ data for a period longer than 30 minutes were then classified as dust event days (DED).

To better understand the overall aerosol composition at the study site location, shorter wavelengths were then examined to classify the other aerosol pollutants. Fine-mode aerosols are more accurately recorded at short wavelengths to get accurately observed optical thickness (Dubovik et al., 2002) and are used here to examine and define these aerosol types (Table 1; Dubovik et al., 2002; Verma et al., 2015). The AERONET retrieval algorithm uses AOD, and sky radiance measurements are input to determine volume particle size distributions at 22 logarithmically equidistant discrete effective radius (r) sizes from 0.05 μm ≤ r ≤15 μm (Dubovik et al., 2000), which are used to estimate particle diameters. These data allow some calculation of the aggregated range of aerosol diameters observed at the site, and their relative frequency (Dubovick et al., 2002). In addition, given that α values of > 2.0 indicates fine mode and lower values indicate coarse mode (Santese et al., 2007; Schuster et al., 2006), these data can also be used to explore how dust properties change over the duration of an event, and how they differ between sites. Single scattering albedo (SSA) data were also derived from the AERONET inversion





products. SSA provides the ratio of scattering and extinction coefficients, and is dependent upon concentration of particles, shape, and size distribution (Jacobson, 2000; Yu et al., 2013).

Particle size is an important indicator to help understand what wavelength should be selected to detect coarse-mode dust in AERONET data (Dubovik et al., 2002; Eck et al., 2010). Other polluting aerosols are fine-mode, and therefore shorter wavelengths are used for their detection (Eck et al., 2010). To identify dust in combination with other aerosols these shorter wavelengths must be used. Typical aerosol definitions were used for 500 nm wavelength data (e.g., Verma et al., 2015) and then amalgamated with typical desert dust thresholds that have been previously applied at 1020 nm (Dubovik et al., 2002). Dust events are best recorded at the longer 1020 nm wavelength as non-spherical dust grains cause scattering that give artificial spectral dependence of the real part of the refractive index at shorter wavelengths, allowing for more robust regression of optical parameters at longer wavelengths (Dubovik et al., 2002).

**Table 1. Aerosol type and the threshold used to classify them at the 500 nm wavelength from Verma et al. (2015) and 1020 nm wavelength from Dubovik et al. (2002).**

| | Thresholds from Verma et al. (2015) | | Thresholds from Dubovik et al. (2002) | |
|---|---|---|---|---|
| **Aerosol Type** | AOD (500 nm) | Angström exponent (470-800nm) | AOD (1020 nm) | Angström exponent (470 - 800 nm) |
| **Marine** | <0.4 | <0.4 | n/a | n/a |
| **Industrial/ biomass burning** | >0.45 | >1.2 | n/a | n/a |
| **Desert dust** | >0.45 | <0.4 | >0.3 | <0.6 |
| **Arid background** | - | 0.4< & <1.0 | n/a | n/a |
| **Mixed type** | All non-defined | All non-defined | All non-defined | All non-defined |

### 3.3 Remote Sensing Data

### 3.3.1 Remote Cameras (RC)

As noted, dust events are not always recorded by satellite remote sensing systems due to issues related to the low magnitude and discrete space and time scale of dust emission and advection (Murray et al., 2016; Bryant and Baddock, 2021), or bias associated with the cloud cover and the satellite overpass time; both being especially important for the observation of HLMA (e.g., Bullard et al., 2016). Urban et al. (2018) refined this understanding further using remote cameras (RC) with images taken of a low latitude desert dust source every 0.25 h during



daylight between 2010 and 2016, yielding a time series of 135,000 images. Despite reporting that dust emissions occurred on 20% of the camera-sampled days, Urban et al. (2018) were able to confirm from (i) examination of MODIS satellite images on the same days and (ii) ground observations (e.g., visibility), that dust was not observed by either of those techniques. Evan (2019) develops the RC approach further to combine field PM10 (particulate

matter less than 10 μm) measurements and a time-lapse RC (0.3 h sample) with a 360° FOV and was able to monitor and quantify fugitive dust emissions in the Salton Sea, California. In our study, to determine the presence of dust events, oblique RCs were set up at various locations across the Á'áy Chù delta to capture a range of dust event information (figure 3; dust event time, source location, dust advection trajectory). The dust plume origin, height, and frequency data collected by RCs operated at a space/time scale that facilitated direct comparison with

synergistic ground (e.g., AERONET, meteorological data) and space-based (e.g., LEO remote sensing) observations. Two cameras were used in the study, one directed down-valley from the former island located at the mouth of the delta and second just off the Alaskan highway to the south of the delta (refer to figure 3). These RCs recorded images of the study site every 10 minutes over a full 24-hour period each day and allowed us to generate a complete chronology of dust emissions for specific time periods within the study period (covering the dust

seasons of 2018 and 2019).





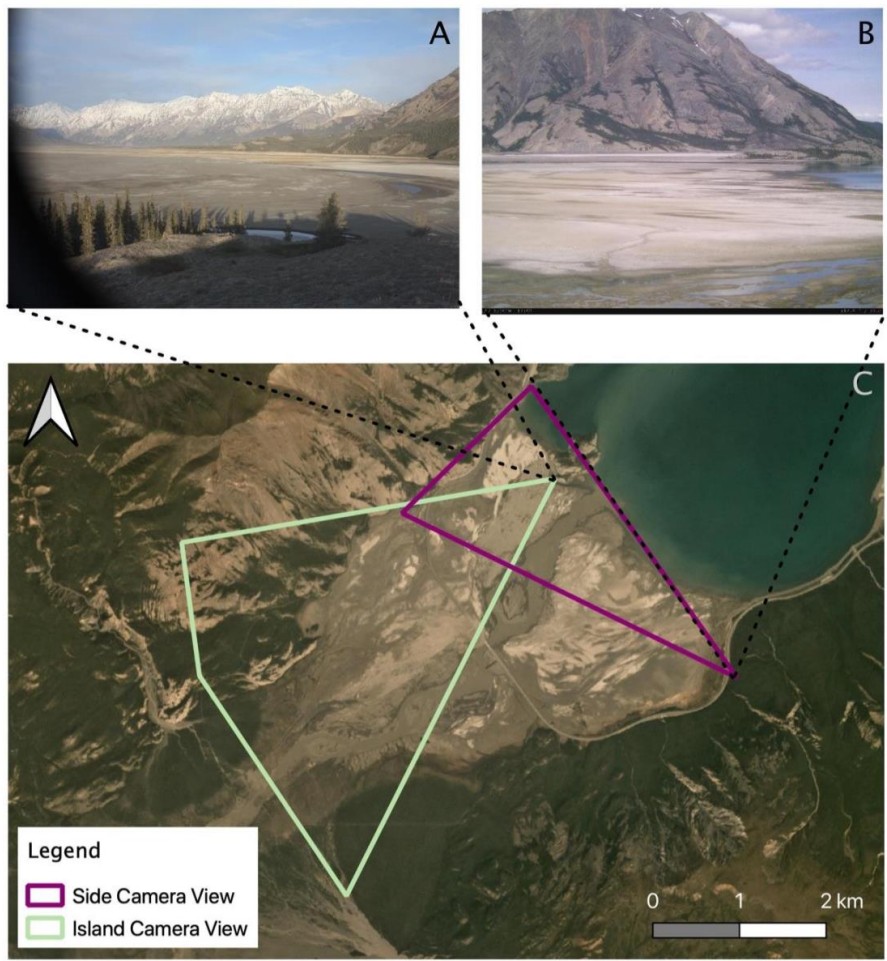

**Figure 3. Locations of the oblique cameras and their approximate fields of view (3C). The Island camera in green is located on the former island in the delta looking south-west (3A). The Side camera is located near the Alaskan Highway looking north-west. Images so dust free views of the cameras (3B). Base imagery is from Planet imagery in June 2018.**

### 3.3.1 PlanetScope Data (LEO)

At the time of the study Planet Labs operated approximately 120 CubeSat satellites in orbit which cover the whole globe every day (Foster et al., 2016; Baddock et al., 2021). The constellation of satellites used in this study are the PlanetScope satellites which have a spatial resolution of around 3m. PlanetScope imagery provides high resolution 4-band (red, green, blue, and near infra-red (NIR)) images of the study site, with overpass time often occurring near to solar noon. Recently, PlanetScope imagery has proved a useful tool for identifying dust plume origins at a scale comparable to field measurement (Baddock et al., 2021). Here, cloud-free PlanetScope images



were examined to look for evidence of dust activity, from which eight dust images were identified from five DEDs as on some days there were multiple overpasses. Methods described by Baddock et al., (2021) were used to track

dust advection and to determine point dust sources (PDS).

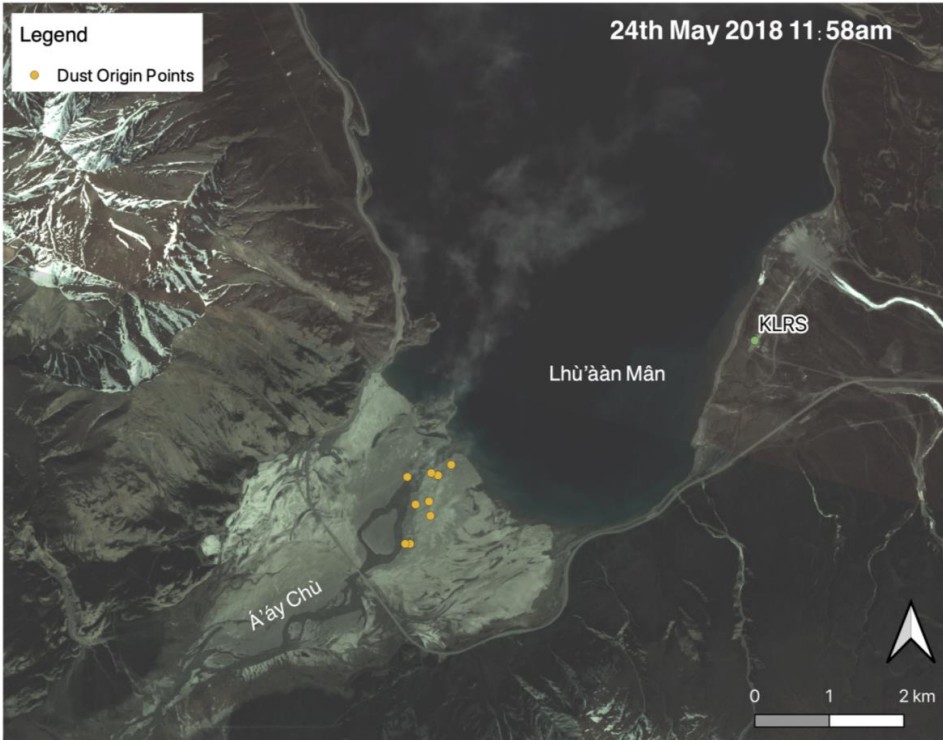

**Figure 4. Image shows the plume rising from the Á'áy Chù delta and going out across the lake captured by PlanetScope (24th May 2018 at 11:58 am local time) overlaid point dust source locations (PDS). Due to the high resolution of the PlanetScope imagery (c. 3 m) we were able to trace the plume to the up-valley source**

**on the delta. The Kluane Lake Research Station's position is identified (KLRS).**

### 3.3.2 MODIS Data (LEO)

Data from the MODIS (Moderate Resolution Imaging Spectrometer) aerosol algorithm Multiangle Implementation of Atmospheric Correction (MAIAC) were used to obtain AOD at 470 nm observations at 1 km spatial resolution for the study site (Lyapustin et al., 2018). The MODIS sensor routinely derives AOD onboard

the Terra and Aqua polar-orbiting platforms. The MAIAC collection 6 processing algorithm (Lyapustin et al., 2018) used in this study divides the globe into fixed 1 km grids and then uses a two-stage combination of time series analysis and pixel/image-based processing to enhance differentiation between background conditions and dynamic atmospheric constituents such as clouds and aerosols (Mharwish et al., 2019). Areas where high AOD gradients exist (e.g., where smoke or dust are both apparent) would normally be at risk removal by the time-series

component algorithm (Lyapustin et al., 2012). To prevent this, and to accommodate for changes in aerosols properties attributed to dust, the MAIAC algorithm deploys a dust test and model to extract AOD in typical 'dusty'



regions (Lyapustin et al., 2018). However, as high latitude regions are not flagged as dust regions, the dust test is not conducted at Lhù'ààn Mân, possibly increasing uncertainty in AOD retrievals where cloud and other aerosols are present. MAIAC AOD from blue (470 nm) and green (550 nm) bands were collected from each of the MODIS overpasses on each day, along with AOD uncertainty based on the blue-band surface brightness. The relationship between daily average AOD at the 470 nm wavelength and AERONET daily average AOD at 500 nm was tested using the Spearman's rank correlation coefficient.

**Table 2. Spectral bands and data quality of spectral data used in this study**

| Spectral Data Used | Wavelength | AERONET Data Quality Level | Application |
|---|---|---|---|
| AERONET AOD (AOD$_D$) | 1020 nm | 1 | Determination of DEDs |
| AERONET AOD (AOD$_A$) | 500 nm | 1 | Comparison against other aerosol types in air column<br><br>Comparison against MODIS MAIAC |
| AERONET Angström exponent (α) | 440-870 nm | 1 | Determination of DEDs |
| AERONET SSA | 440, 675, 870, and 1020 nm | 2 | Radiation scattering effectiveness of aerosols |
| AERONET Volume Size Distribution | 340, 380, 440, 500, 675, 870, 1020, and 1640 nm | 2 | The percentage of spherical particles in the observed aerosol to determine peaks in particle size |
| MODIS MAIAC Algorithm (AOD$_M$) | 470 nm | n/a | Space-based AOD estimates |

## 4. Results and Discussion

### 4.1 Event-Scale Observations

*AERONET AOD retrieval*

By combining the oblique camera images, AERONET, and meteorological data, a detailed picture of dust activity and detection at the site can be drawn. Relationships between AOD or α and various meteorological factors on the



24th of May 2018 are examined at an event scale in Figure 5. Two dust events were recorded by the AERONET station on this day, one in the morning from 07:30 to 09:30 and another in the afternoon at 14:30 to 18:30 local time. Maximum wind speed and $AOD_D$ do not increase in tandem during the dust events, with the two factors observing a negative relationship with a Spearman's Ranks coefficient of -0.296. In the morning, the event starts

at a maximum wind speed of 3.7 m s$^{-1}$ which is below typical initiation thresholds. A large increase in wind maximum speed occurs after 10:00 and is sustained above 6 m s$^{-1}$ from 12:00 to 20:00. The dust event in the afternoon and subsequent increase in $AOD_D$ does not start until 14:30, four hours after the increase in wind speed, in these initial four hours $AOD_D$ decreases and remains below 0.25 for this time. Finer particles which had limited effect on the $AOD_D$ are initially picked up during the increase in wind speed, evidenced by panel B where α

remains consistently high fluctuating between 0.25 and 0.75 until the start of the dust event at 14:30. Wind direction (Figure 5F) changes abruptly at 11:30 to be orientated from 230-260°. This shift coincides with a steep increase in wind speed. In the afternoon, the wind direction is consistently coming from the direction of the delta (230-260°) indicating topographically forced drainage winds, before shifting back to the south-east in the night. Both dramatic drops in wind speed at 09:00 and 19:30 coincide with periods of no dust being observed in the

oblique camera images.

Relative humidity and temperature seem to follow no distinct pattern in α and $AOD_D$. The relative humidity follows a diurnal pattern of peaking just after sunrise at 60 %, decreasing to 18.8 % late afternoon, and then increasing again at dusk. On 24th May 2018, the start of the increase of relative humidity coincided with the end of the afternoon dust event. Temperature follows the opposite cycle of the relative humidity with temperatures

being at their lowest at 06:00 at 0.9°C, then increasing through the day peaking at 14.4°C at 18:30. A steep increase in temperature coincides with the change in wind speed and direction at 11:30.

From the island camera, a large dust event was detected at 13:30, where a high and dense dust plume can be seen crossing the island, resulting in poor visibility. In Figure 6, images taken at the start and subsequent peak of dust events detected by the AERONET station were used to check if plume orientation affected detection. At 07:40,

dust is observed further south along the delta than at other times in the day, a southern movement is not detected on the afternoon event.





**Figure 5. Aerosol spectral parameters and meteorological conditions for the 24th May 2018. Dust events defined by the Dubovic et al. (2002) threshold are highlighted in grey, dust events evident on the oblique imagery are highlighted in pink. Panel (A) is AOD_D, panel (B) is DQ 1.0 Angström exponent (α) at 440-870nm, panel (C) is Relative Humidity (%), panel (D) is half hourly pressure (mbar), panel (E) is half hourly temperature (°C) and panel (F) is maximum wind speed (m s⁻¹) and direction (°). There are no AERONET AOD returns in DQ 1.5 and 2.0 for this day.**



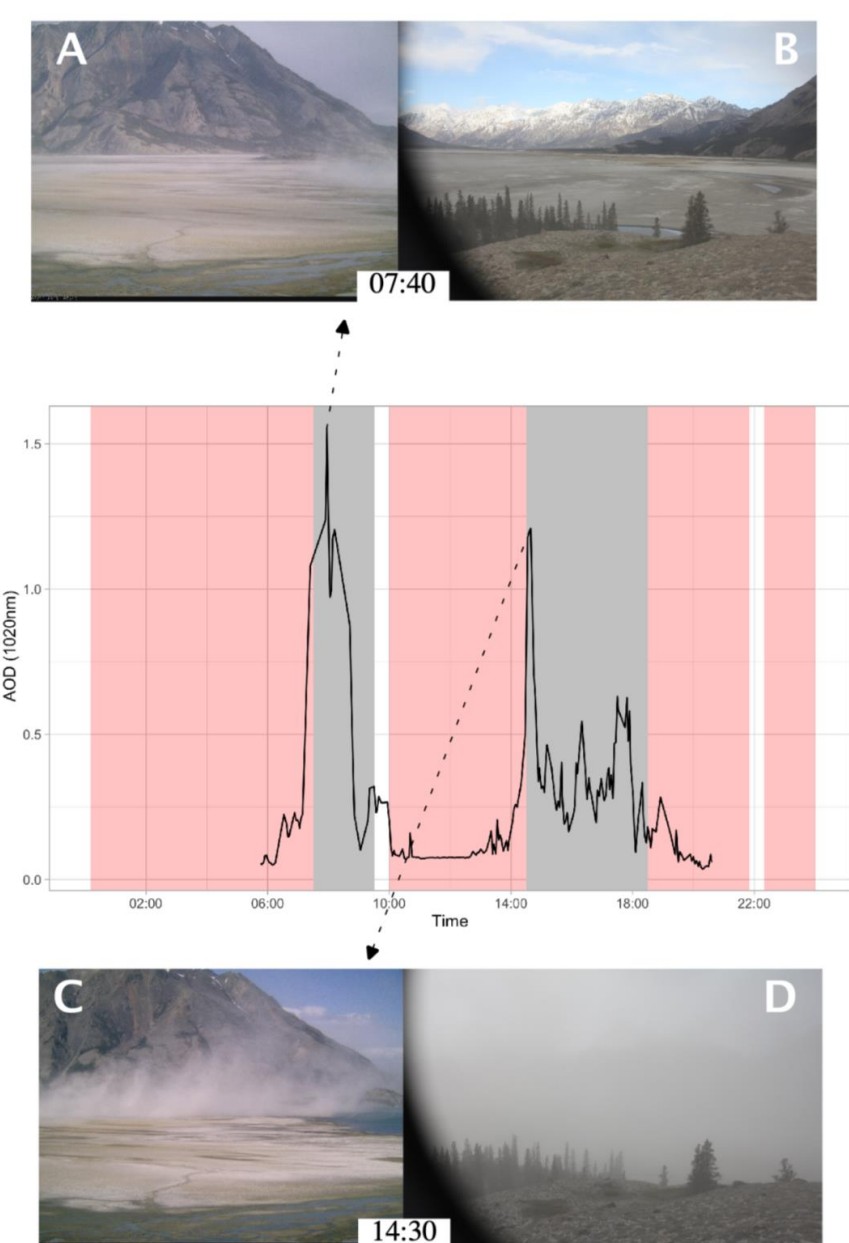


**Figure 6. AOD$_D$ returns from the 24/05/2018 visualised with the corresponding oblique camera images during peak events. In the morning dust is in the southern section of delta.**

Throughout both AERONET detected dust events on the 24th of May 2018 the AOD$_D$ fluctuates. The start of both events is signified by steep increases in AOD followed by significant decreases. In the morning event, AOD$_D$

dropped from 1.2 to a considerably lower 0.32, and the afternoon event started at 0.49, decreasing to 0.31 after 30



minutes and to then peak at 0.63 one hour later. When wind is above threshold, dry surface sediments are quickly entrained, allowing for the peak in $AOD_D$ at the start of dust events. Once the surface layer of sediment has been removed through aeolian processes, damp sediment is exposed and temporarily limits sediment availability (Chepil, 1956), resulting in the drop in $AOD_D$. Over time, surface winds that are above entrainment threshold

(Figure 5F) coupled with high temperatures (Figure 5E) work to dry these sediments and increasing the availability for transport (Cornelis and Gabriels, 2003). These processes were observed at the study site by Nickling (1978) who concluded that the dynamic relationship between wind and sediment moisture resulted in highly periodic dust emissions at the site, where the wetting and drying cycle controls the wind erosion process (Wang et al., 2014). Complex microtopography of the delta evident in the RC images (Figures 6 and 7), also leads to heterogeneity

and variability in sediment drying rates. This process accounts for localised resumption of dust emissions and fluctuations in AOD observed here (Ravi et al., 2006).

When investigating similar dust events at the study site across the 2018 measurement period, a significant positive relationship (Spearman's Rank > 0.8) between $AOD_D$ and maximum wind speed was not observed, suggesting that wind speed is not the only factor that affected dust emission. Several factors may account for the lack of

relationship between the wind speed and changes in both $AOD_D$ and α including soil moisture, relative humidity, and air turbulence. Nickling (1978) observed at the Á'áy Chù delta that suspended sediment transport rates are more closely related to air turbulence than wind velocity. The meteorological station used in this study is located 4 kilometres away from the dust source, and wind velocity is measured at one height (2 m above the ground), so measures of turbulence are not routinely quantified. The distance from source and complex wind regimes at the

study site may mean that local discrete winds and relative humidity fluctuations are not observed by the KLRS meteorological station. Without use of a higher density of observations along the valley to characterise dust source wind velocity, turbulence, and relative humidity it is impossible to determine definitive wind threshold controls on dust emissions at the site.

*Dust within the AERONET FOV*

Even with high-quality AERONET data within kilometres from the dust source, it is evident from RC data that dust events may not always be detected within the FOV of the AERONET station. When the RC data are used in conjunction with the high-resolution PlanetScope imagery (see figure 4), the location of plume origins and plume advection allows AERONET station FOV detection, and overall rates to be characterised.

Several factors may explain why events are not being captured. The side camera images display that when dust is

detected by AERONET in the morning, the dust is in motion more on the southern side of the delta, possibly suggesting that to be observed by the AERONET station, dust emissions need to originate in the southern sector of the delta. Southern emissions of dust were not observed in the afternoon event. In fact, most plume origins from 2018 were mapped to be in the central to northern section of the delta (Figure 4). Once sediment is suspended, katabatic winds blow it out across the lake, where it meets the valley wind and is blown north north-west away

from the AERONET station. This dust does not pass directly over the sensor and may also remain relatively low and below the detection arc of the sun-photometer. Sun-photometers use direct sun measurements to determine AOD and α (Holben et al., 1998). However, in mountainous terrain if dust remains below the mountain line, the photometer will be unlikely to record it. At this time of year oblique camera images (Figure 7) can observe dust



in suspension for an extended period after the sun has gone below the horizon. This is not recorded by the sun-photometer. A possible explanation as to why AERONET detected dust in the afternoon when the plume was not originating in the southern section of the delta is that the vertical motion of emitted dust was greater so that it was above the mountain line and thus detectable by AERONET. A combination of dust emission at night and low height of the dust lead to dust event's detection being underrepresented at Lhù'ààn Mân

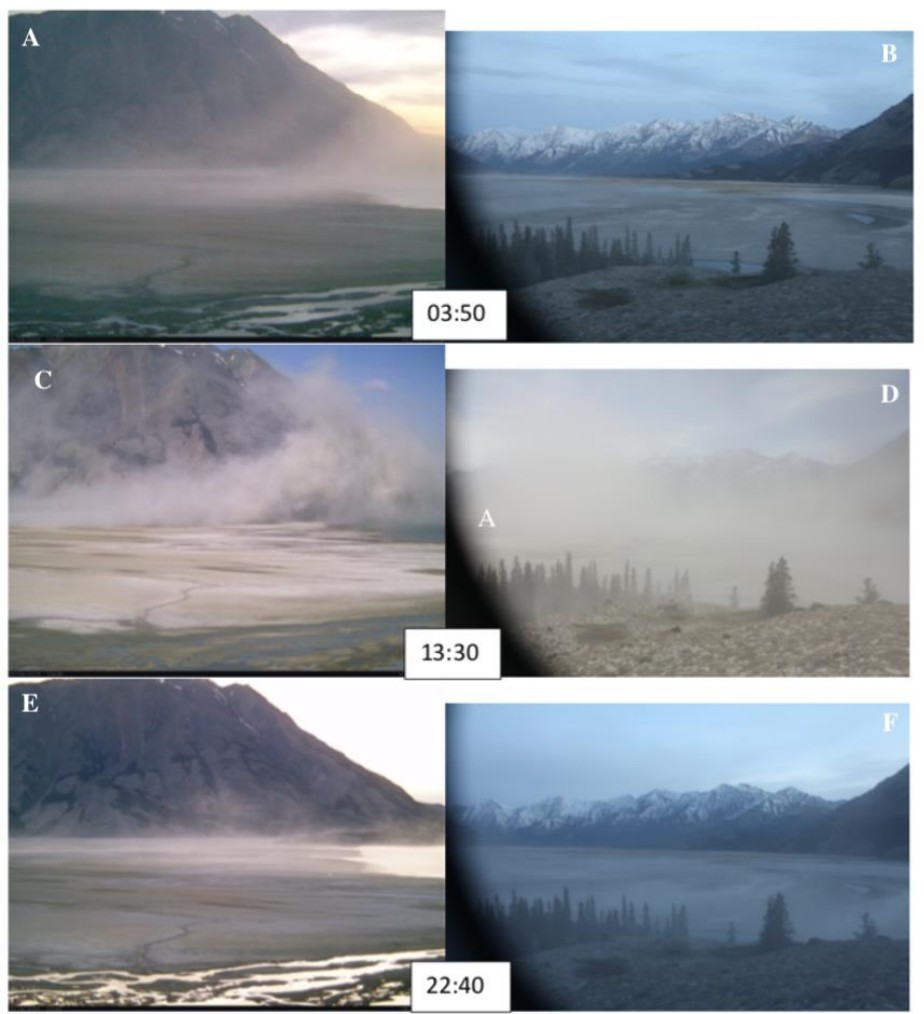

**Figure 7. Oblique Camera images from selected times of dust events on the 24th May 2018. These images were all captured when dust is not detected at the AERONET station but reveal that dust is present at the site and is not detected by AERONET. Locations of cameras are detailed in Figure 3. On the left column images are taken from the side camera and the right column images are taken from the island camera.**



**4.2 Controls on dust emissions**

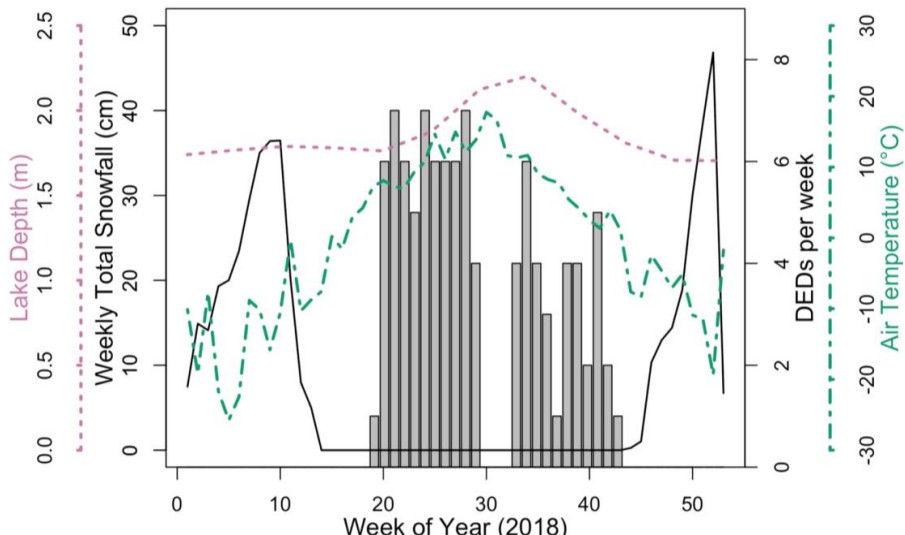

**Figure 8. Variability in DEDs in 2018 and selected seasonal variables that affect dust emission. DED is the total number of dust event days recorded by AERONET from AOD$_D$ in the given week. The AERONET**
**station was recording from 14/05/18 until 21/10/18, but no data was recorded between 24/07/18 - 14/08/18. Average weekly depth (cm) displayed with the solid line and air temperature (°C) displayed with the dot-dash line recorded at the Lhù'ààn Mân Research Site. Lake height displayed with the long dash line is taken from the Environment Canada lake depth gauge 09CA001 at Kluane Lake near Burwash Landing and is the average weekly water depth at that site.**


The occurrence of dust events in this region reflects the relationships between sediment supply and availability, and the co-dependency between aeolian, fluvial, and glacial systems. Average air temperature, snow depth and lake depth displayed in figure 8, all affect the supply or availability of sediment entrainment in the valley. Over the winter, cold air temperatures result in very low discharge from the Kaskawulsh glacier and snow covering the
ground inhibits dust entrainment as the layer of snow protects the underlying sediment from the wind (Bullard, 2013; Bullard et al., 2016). In April, snow cover begins to rapidly decrease, and with this, dust events are expected to increase as seen in other high-latitude locations (Bullard et al., 2016). This is not reflected at Lhù'ààn Mân in 2018. The AERONET station began recording on 13th May 2018 when dust was recorded at the site. With no aerosol records for April, dust events cannot be determined. Consequently, 13th May was most likely not the first
DED. Any lack of events in April can be attributed to several factors; the average air temperature in the month of April is still -1.51 °C, snow cover is still decreasing over this month and melted snow refreezing in the ground, increases interparticle cohesion and cementing the particles together (Wang et al., 2014). Dust events are pronounced in May, when the average air temperature is 6.82 °C and the soil over the delta has thawed and the mostly dry, loose surface soil is easily deflated. Through the freeze/thawing process, a degree of sediment sorting
occurs whereby finer sediments are moved to the surface (Corte, 1971; Bateman, 2013). These finer particles are more readily entrained by the wind as the smaller diameter requires a lower critical shear velocity, and the finer





grains are no longer shielded by coarser particles. Sorting is more pronounced after several freeze/thaw cycles. The frequency of dust events decreases in October, when the average air temperature is 1.61 °C, and the ground begins to freeze again. The decrease in DEDs in October can also be attributed to the removal of the AERONET

station during this month.

The diversion occurring in June 2016 (Shugar et al., 2017), resulted in the highly dynamic braided Á'áy Chù becoming incised, with the main river channel having not changed position since 2017. Sediment is therefore available for entrainment through the entire summer season, as at peak discharge the floodplain does not become inundated with meltwater. Other high-latitude locations observe two distinct peaks in dust emission over the melt

season, one early season when the snow has melted and one later season when flow is again reduced (Bullard and Mockford, 2018; Crusius et al., 2011). Lhù'ààn Mân differs from traditional high-latitude regions' seasonality in dust emissions as it does not experience the reduction in dust events coinciding with flooding associated with peak river discharge. The delta is no longer inundated with meltwater, sediment is no longer replenished on a yearly scale. Dust is emitted again at the start of the following season due to freeze/thaw action bringing smaller grains

to the surface (Corte, 1971; Bateman, 2013). Recolonisation of vegetation on the floodplain may increase surface roughness, decreasing dust emissions at the site into the future.

### 4.3 AERONET Aerosol Retrievals and a DED time series

Cloud-screening is an essential aspect of the AERONET network data refinement, as clouds affect AOD, with cirrus clouds contributing to fine-mode AOD and other cloud types impacting coarse-mode retrievals (Arola et

al., 2016). Cloud screening is applied to all AERONET data to remove any cloud related uncertainties in AOD retrievals at all wavelengths (Arola et al., 2016). However, at our study site, this approach removes most of the data relating to either known or observable dust fluxes at the study site. As noted in Figure 9a, the number of possible AERONET DEDs recorded in 2018 and 2019 using the definition at each DQ level varies significantly. The initial DQ 2 algorithm inevitably excludes the vast majority (97.8%) of retrievals, and a distinct decrease in

number of possible DEDs are also observed as processing steps between DQ 1.0 and level 1.5; with eight months where no DED are recorded at DQ 1.5 and 2.0. Figure 9a also displays the annual cycle of DQ 1.0 DEDs at Lhù'ààn Mân, with DEDs occurring through the recording months. June in both 2018 and 2019 produced the highest frequency of DEDs at 26 and 20 respectively.

Nevertheless, where DQ 1.0, 1.5, and 2.0 data are applied, it remains likely that some cloud optical depth may also be included in the spectral AOD measurements. However, Arola et al. (2016) found that uncloud-screened DQ data added roughly 0.007 and 0.0012 onto the AOD values. To investigate aerosol types at out study site, we applied a combined approach, whereby DQ 1.0 data were used to build an inventory of DEDs for comparison with in situ data, and SSA and volume size distribution inversion products were generated using DQ 2.0 cloud-screened

data. The average $AOD_D$ at Lhù'ààn Mân for the 2018 measurement season was 0.181, 0.031, 0.03 for DQ 1.0, 1.5, and 2.0 prospectively which depicts an almost pristine atmosphere (also see figure 10c&d). However, Figure 9b, which derives aerosol types from the DQ 1.0 data using the method of Verma et al. (2015), also highlights the possible presence of a range of aerosol types at this site. From these data we suggest that the two dominant aerosols



that alter the incoming solar radiation at this location are dust and biomass burning particles, where dust accounts
for 11.4% of daily aerosol occurrences at the site.

Given the distribution of monitoring sites, most previous studies which investigate the radiative signature of dust
using AERONET returns have been conducted at low latitudes (desert dust) using DQ 1.5 AERONET data (e.g.,
Santese et al., 2013; Binietoglou et al., 2015). In this study, the likely presence of dust events was determined
through use of initial generic thresholds at two different AERONET wavelengths, 500 nm and 1020 nm. When
compared to direct ground data observations, AERONET derived dust events in this study recorded at longer
wavelengths were found to be a closer match to the known frequency of events than those at shorter wavelengths
(Figure 9ab), and we therefore note that HLMA determined using shorter wavelengths it may lead to an under-
representation of dust event and DED frequency. For example, on a day where RC data shows dust events for
95.8% of the day (24th May 2018), 11.6% AOD readings were classified as dust using the thresholds from Verma
et al. (2015), whereas thresholds from Dubovik et al. (2002) yielded 24.2% AOD readings as dust.

When investigating AERONET retrievals at this location, consideration of differences in the optical properties of
glacially derived dust when compared to those derived in the mid-latitudes are important. Mid-latitude AOD and
$\alpha$ thresholds (from Verma et al., 2015 and Dubovic et al., 2002) have been used to define dust presence in this
study. However, these thresholds may not encompass the optical parameters of HLMA and this may also impact
retrievals at this site.

For DEDs recorded at Lhù'ààn Mân in 2018 using definitions from Dubovik *et al.* (2002) at DQ 1.0, the average
$\alpha$ was -0.003 indicating a very coarse grain size. The average $\alpha$ increased in 2019 to 0.12, suggesting a slight
decrease in grain size. Both the $\alpha$ and $AOD_D$ for DEDs at Lhù'ààn Mân suggest a predominantly unimodal grain
size distribution (Figure 11c) suggesting that grains in suspension have comparable size and optical properties.
Figure 11a outlines AERONET inversion volume size distributions for DEDs at the study site. The study site is
dominated by the presence of biomass burning aerosols (smoke). Scans made on DEDs (Figure 11a) show a
bimodal distribution with peaks at 2.6 and 10.1μm. Bachelder et al. (2020) calculated the normalised particle size
distribution through deposition traps and optical particle counters at Lhù'ààn Mân and recorded a maximum
frequency at a particle diameter of 3.25 μm. This is coarser than the first maximum frequency observed at 2.6 μm
from the first peak and significantly finer than the 10.1 μm second peak calculated from the AERONET inversion
products in this study. The second peak observed in the volume size distributions are similar to studies of well
characterised dust sources in the mid-latitudes which peak around 10 μm (Kok et al., 2017; Huang et al., 2018).
SSA was derived for the aerosol phases show that dust scatter the most incoming radiation with biomass burning
aerosols scattering slightly less (figure 11c).




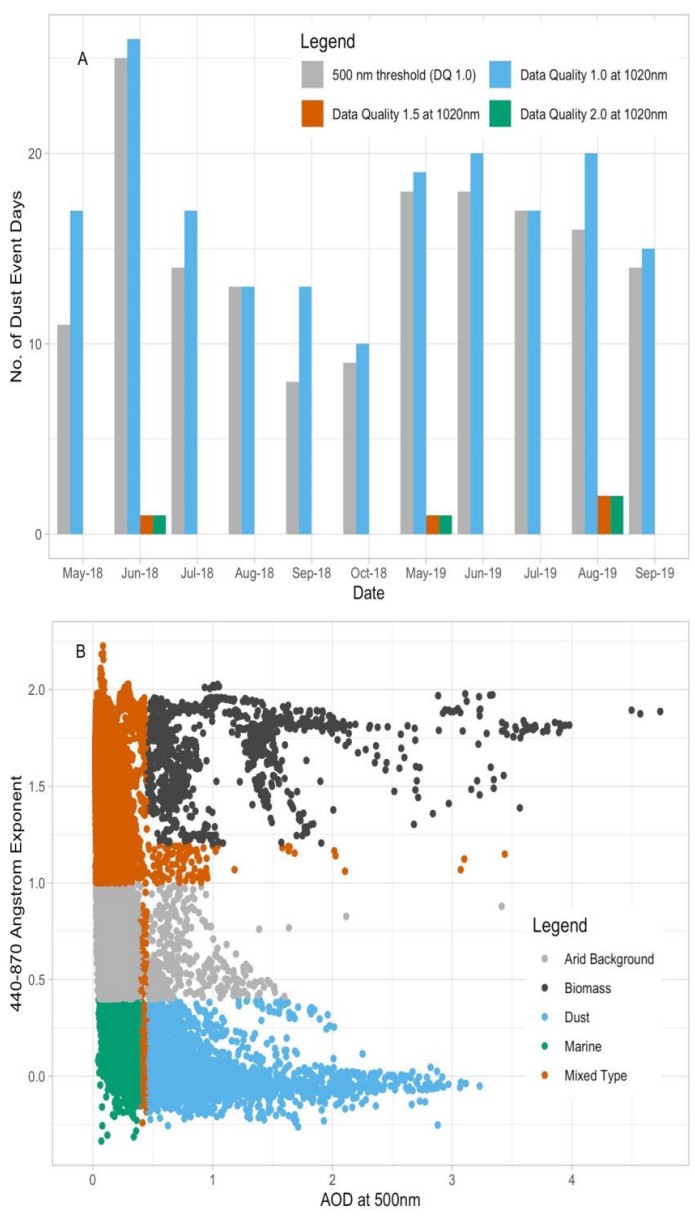

**Figure 9. (a)** **The number of DEDs by month from 1020 nm over the 2018 and 2019 Lhù'ààn Mân dust seasons detected by the different data quality levels computed by AERONET. DQ 1.0 is represented in blue, DQ 1.5 is represented in orange, and DQ 2.0 in green. (b) Scatterplot of daily DQ 1.0 AOD$_A$ and α at 440-870nm used to classify aerosols at Lhù'ààn Mân for DEDs in the dust seasons of 2018 and 2019. In yellow is the dust, black is biomass/industrial aerosols, grey is the arid background, blue is marine aerosols and mixed types are represented in pink. Thresholds for each class are taken from Verma et al. (2015).**






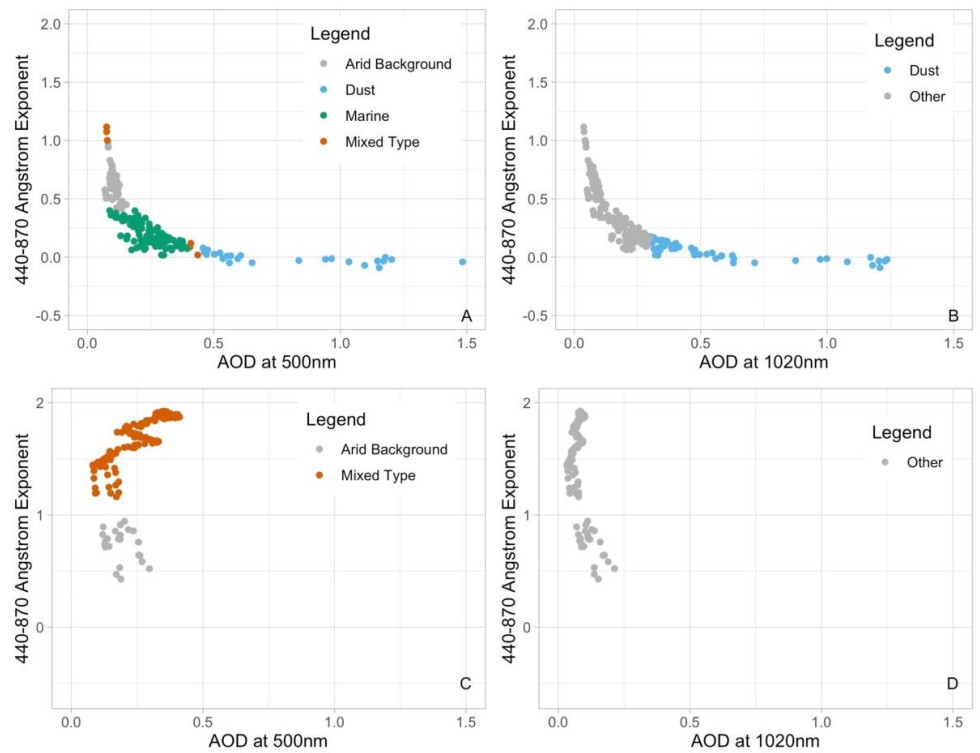


**Figure 10. Aerosol compositions comparisons at Lhù'ààn Mân on a DED - 24/05/2018 (A&B) and a non-dust event day- 23/07/2018 (C&D). Scatterplots A&C are at DQ 1.0 AOD at 500 nm with thresholds from Verma et al. (2015) and scatterplots B&D are at DQ 1.0 AOD$_D$ with thresholds from Dubovik et al. (2002). In yellow is the dust, black is biomass/industrial aerosols, grey is the arid background, blue is marine**
**aerosols and mixed types are represented in pink.**



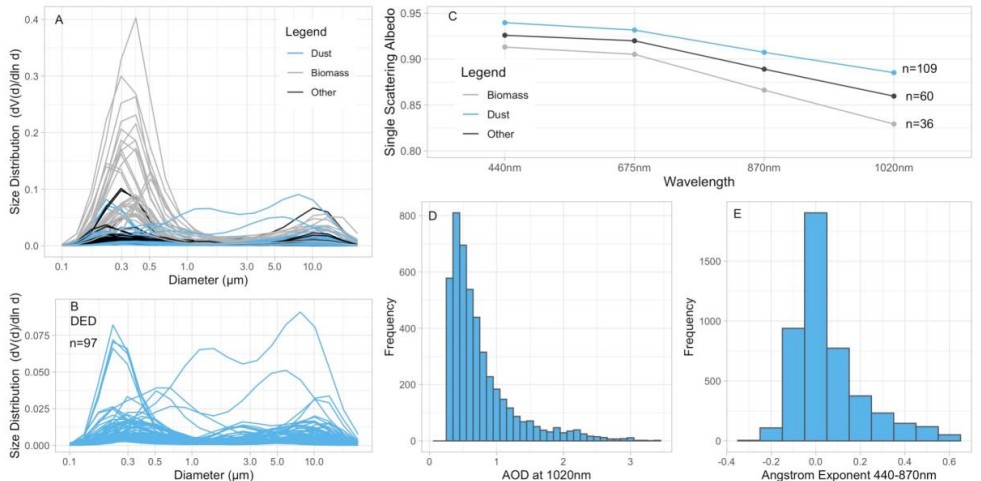

**Figure 11. (a) Volume size distribution of aerosols at Lhù'ààn Mân Recordings that were taken on a DED (DED definition from Dubovik et al. (2002) in the 1020 nm band) are in pink, and recordings taken when there were biomass burning (defined from Verma et al., 2015) events (smoke) are in blue. Any days where dust or smoke were not detected in AERONET are displayed in black. Models used to calculate volume size distribution are based on AERONET level 2.0 DQ. Dust event days are then pulled out for (b) underneath to show the presence of bimodal dust size. (c) Single scattering albedo for the principal aerosol types recorded at the Lhù'ààn Mân AERONET station for 2019. Dust events are in green, biomass burning events are in pink, all other aerosol types are in blue. Models that calculate SSA are based on AERONET level 2.0 DQ. Frequency histograms of AOD (d) and Angstrom exponent (e) at Kluane Lake AERONET station over 2018 and 2019.**

### 4.3 LEO-based observations

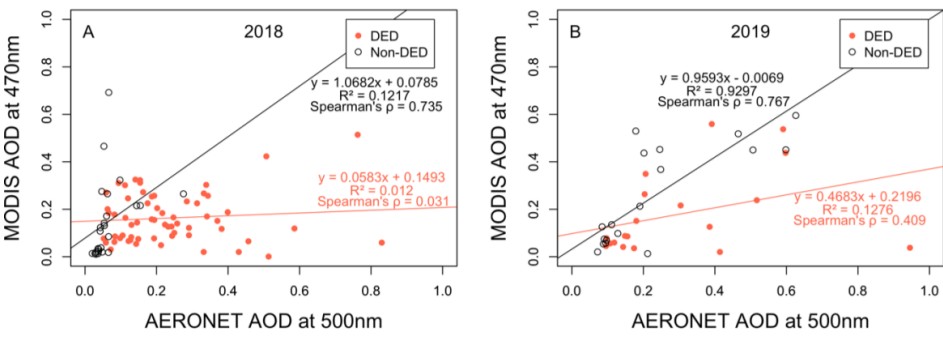

**Figure 12. Evaluation of daily mean MAIAC v6 1km AOD at 470 nm against daily AERONET mean AOD at 500 nm at Lhù'ààn Mân over 2018 (A) and 2019 (B). Note that MAIAC operates using AOD at two wavelengths, 470 nm and 550 nm. 470 nm was chosen due to its relative closeness AERONET AOD at**



**500nm. DEDs are visualised in red and non-event days are in black. Regression lines equations, R$^2$ values,**
**and Spearman's rho for the different day types are also provided.**

MODIS MAIAC data were analysed to establish dust event occurrence, typical associated AOD$_M$ returns, and
gross comparison with synergistic AERONET AOD$_A$ data at the study site. Differences in retrievals between
DEDs and non-event days were recorded (see figure 12). The relationship between AERONET and MODIS AOD
on DEDs is weak (Spearman's rho value of 0.031 and 0.409 in 2018 and 2019 respectively); many events are
recorded via AOD$_A$ but matched with AOD$_M$ are relatively limited. Non-dust event days observe a stronger
association (Spearman's rho value of 0.735 in 2018 and 0.767 in 2019).

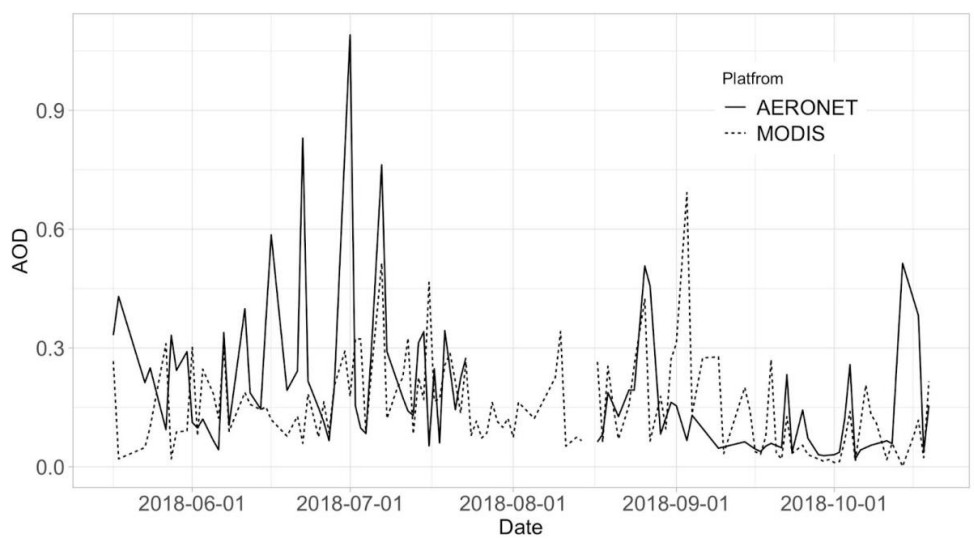

**Figure 13. Mean daily MAIAC v6 1km AOD at 470 nm (dashed line) and mean daily AERONET AOD at**
**500 nm (solid line) returns from 17/05/18 – 17/10/18 on days where both platforms were recording. Panel**
**A shows the maximum AOD value recorded that day by each platform.**

Figures 8 and 9 suggest a weak relationship between AOD$_A$ and AOD$_M$ at Lhù'ààn Mân during 2018. Previous
studies using similar data but based in the mid-latitudes, have shown significantly stronger correlation between
these variables over similar time intervals (e.g., Jethva et al., 2019; Martins et al., 2017; Mhawish et al., 2019).
Analysis of AOD$_M$ and AOD$_A$ data in this study is constrained by the retrieval wavelengths used to obtain AOD.
The MODIS MAIAC algorithm derives AOD at both 470 and 550 nm, the closest AERONET wavelength is 500
nm. While the difference in wavelengths may increase uncertainty in the relationship between the two detection
approaches, it is clear from AERONET data that cloud-screening quality control are likely to be more important
confounding issues. However, because of the nature of how MAIAC and similar MODIS aerosol retrievals are
generated, uncloud-screened AOD are not available. In this instance, the MAIAC collection 6 processing
algorithm uses a background time series to identify atmospheric conditions that change quickly (e.g., to capture
clouds or dust). As noted, a dust test is conducted at low latitudes to differentiate dust, so that this aerosol species



is not systematically excluded from the data (Lyapustin et al., 2018). However, the region containing Lhù'ààn
Mân is not in this dust test, and consequently dust events that are active at the time of overpass are often excluded.
Indeed, maximum $AOD_A$ values observed by uncloud-screened AERONET data during dust events are often
missing in the MAIAC $AOD_M$ data (Figure 13). We therefore suggest that systematic removal of dust events in
MAIAC and other similar LEO AOD retrievals accounts for the large discrepancies in $AOD_A$ and $AOD_M$ for

DEDs here (Figure 12).

### 4. Conclusions

It is evident from this study that high-latitude dust is being fundamentally under-represented when using both
ground and space remote sensing methods and therefore in both past and present global dust and climate models.
Understanding the effect that high-latitude dust has on the planet is extremely important as we move forward into

a time of increased climate warming. This effect is calculated using models which often use remote sensing to set
their parameters. Correct parameterisation of model inputs, such as dust grain size, spectral signatures, source
locations, and detection rates are extremely important to ensure that the influence of dust is properly assessed and
that future predictions are accurate. At Lhù'ààn Mân, an exceptional frequent emitter of dust, more than 97.8% of
events are not being detected. This number may be much greater at other high-latitude locations without an

AERONET station and with more discrete dust emissions, for example the Copper River in Alaska. Thresholds
used for the detection of dust events in this study may also not encompass all dust events at the sites, as different
wavelengths return different frequency of events. Oblique camera images provide evidence that most dust events
are being missed by AERONET. On an event scale 73.9% of emissions were missed. Missed events occur possibly
due to a combination of the thresholds for detecting high-latitude dust being incorrect and AERONET detection

issues in mountainous areas and during darkness. For dust to be detected correctly the high-latitude dust threshold
in AERONET data needs to be revisited.

### Data availability

AERONET data is available to download at https://aeronet.gsfc.nasa.gov/cgi-
bin/data_display_aod_v3?site=Kluane_Lake&nachal=2&level=3&place_code=10 . The Burwash landing lake

depth gauge data is available to download at https://wateroffice.ec.gc.ca/report/historical_e.html?stn=09CA001 .
The RC camera and meteorological data can be found here https://doi.org/10.5281/zenodo.7249227.

### Authorship contribution

**R.H.** conceptualisation, methodology, formal analysis, writing - original draft. **R.G.B** conceptualisation,

methodology, writing - review & editing. **J.K**. data collection, methodology, writing - review & editing.

### Declaration of competing interest

The authors declare that they have no conflict of interest

### Acknowledgements

We would also like to acknowledge that the Kluane Lake Research Station is located on the traditional territories
of the Kluane First Nations and the Champagne and Aishihik First Nations. Thanks is also due to Environment



and Climate Change Canada in their effort in establishing and maintaining Kluane Lake AERONET station.
Environment Canada for the use of their data for the lake depth gauge at Burwash Landing. Funding support is
acknowledged for JK by NSERC RGPIN-2016-05417 and from the Canadian Mountain Network. The authors
acknowledge Planet Labs for their provision of free data access to education and research users.

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
