# Peer review of "The (mis)identification of high latitude dust events using remote sensing methods in the Yukon, Canada: A sub daily variability analysis."

_EGUsphere, 2022_

## Author Comment (AC1)

**Author's response to reviewer number 1**

AC: We thank the anonymous referee for the detailed review of our manuscript. We carefully reviewed each comment and have amended at manuscript to address the issues raised. Reviewer comments are in black with our responses in blue. Changes to the manuscript are in the small font size 10.

RC1: General major comment.

While I think that the current title is fine, I believe that it does not highlight some of the more important results from this study. In my opinion, the more important points are two: 1) it unequivocally demonstrates with observations that high latitude dust activity can be very frequent and abundant 2) that existing mainstream instrumentation such as satellite and Aeronet can miss significantly a number of events and demonstrate they are not suitable for a climatological studies. I think these two facts are more relevant and of importance from the view of incorporating HLD in global surveys and modelling efforts.

AC: We thank the reviewer for this comment. To address your concerns we have retitled the paper, "The (mis)identification of high latitude dust events using remote sensing methods in the Yukon, Canada: A sub daily variability analysis". We hope this is satisfactory.

RC1: In addition, this study demonstrates something that was already reported in the Urban et al and Baddock et al (cited) papers where they excellently demonstrate how modern polar satellites very often miss dust activity to the point that it is clearly undercounting a significant amount of events. As a result, global assessments that rely in satellite data are biased towards lower latitudes. This study further contributes to this concept with the novelty that this is a largely unknown dust activity regime at latitudes not considered in the above studies.

AC: Thank you for your valuable comments and suggestions. Please find our answers to the individual specific comments below.

RC1: Overall comments about satellite images. I read this manuscript in a printed version of the paper. All satellite images (except perhaps figure 2) had poor contrast and the darks were too dark and without definition. I can't tell if it was a problem in my printer, but this is a fact you may want to check before final submission. The PDF in the computer screen looked much better than in print.

AC: We thank the reviewer for this comment. We printed off and reviewed the satellite images and deemed the contrast are okay. We hope this is satisfactory.

RC1: Abstract:

It would be desirable to add information of the periods of time (months/years) of the surveys.

Overall the abstract highlights too much the technical aspect of detecting of changes thresholds and does not report a more important fact: dust activity is much more frequent than previously expected and this project has quantified it. So for example, stating here what frequency was measured with the remote cameras and by Aeronet is a very important fact in my opinion.

AC: We thank the reviewer for this comment and agree. We have updated the abstract so that it now reads:

"The observation and quantification of mineral dust fluxes from high-latitude sources remains difficult due to a known paucity of year-round in situ observations and known limitations of satellite remote sensing data (e.g., cloud cover and dust detection). Here we explore the chronology of dust emissions at a known and instrumented high-latitude dust source: Lhù'ààn Mân (Kluane Lake) in Yukon, Canada. At this location we use oblique time-lapse (RC) cameras as a baseline for analysis of aerosol retrievals from in situ metrological data, AERONET, and co-incident MODIS MAIAC to (i) investigate the daily to annual chronology of dust emissions recorded by these instrumental and remote sensing methods (at timescales ranging from minutes to years), and (ii) use data intercomparisons to comment on the principal factors that control the detection of dust in each case.

Lhù'ààn Mân is a prolific mineral dust source; on the 24/05/2018 the RC captured dust in motion throughout the entire day, with the longest dust-free period lasting only 30 minutes. When compared with time series of RC data, optimised AERONET data only manage an overall 26 % detection rate for events (sub day) but 100% detection rate for dust event days (DED) when dust was within the field of view. Here, in this instance, RC and remote sensing data were able to suggest that the low event detection rate was attributed to fundamental variations in dust advection trajectory, dust plume height, and inherent restrictions in sun angle at high-latitudes. Working with a time series of optimised AOD data (covering 2018/2019), we were able to investigate the gross impacts of DQ choice on DED detection at the month/year scale. Relative to ground observations, AERONET's DQ2.0 cloud screening algorithm may remove as much as 97 % of known dust events (3% detection). Finally, when undertaking an AOD comparison for DED and non-DED retrievals, we find that cloud screening of MODIS/AERONET lead to a combined low sample of co-incident dust events, and weak correlations between retrievals. Our results quantify and explain the extent of under-representation of dust in both ground and space remote sensing method; a factor which impacts on the effective calibration and validation of global climate and dust models."

RC1: Figure 1. Some of the stations in easter Patagonia are high latitude and do report both proglacial and depression dust activity so they should be tagged in pink. If I recall correctly the Neuquen, Comodoro Rivadavia and Rio Gallegos sites are such cases .

AC: Thank you for your comment, the original criteria for defining the HLMA stations was that the station had to be within 10 km of a glacier or any ice. This was rethought to add some leniency (to include most stations within $\geq$ 50 °N and $\geq$ 40 °S latitude) and as a result Comodoro, Neuquen, Rio Gallegos, and other AERONET stations were added. The figure has been modified, numbers updated, and projection corrected.

[Figure]

Figure 1. Location of all AERONET stations with stations in the high-latitudes or proglacial areas highlighted in orange. Cryospheric stations account for 48 out of 1655 global AERONET (data from the AREONET website: https://aeronet.gsfc.nasa.gov).

RC1: Figure 2: can you add the location of Burwash landing?

AC: The authors agree that adding Burwash Landing would provide important contextual data. However, this would require us to extend the map another 30kms to the north which may reduce the clarity of this figure in displaying where the instruments are in relation to the dust source area. Burwash Landing can been seen in Figure S1. We hope that this is sufficient.

RC1: Line 202 : The Aeronet … is a FEDERATED network ….

AC: This comment was considered and added to the manuscript. Thank you for the suggestion. The manuscript now reads:

"The AERONET (Aerosol Robotic Network; Holben, 1998) is a federated network of ground-based sun photometers that measure the rate of solar ray extinction in the atmospheric column above the photometer to determine AOD alongside other atmospheric properties."

RC1: Line 209-2010: add year of operation for those months.

AC: Agreed. The manuscript now reads:

"The Kluane Lake AERONET station (see Figure 2 and S1 for location) recorded data from early May to late October in 2018, 2019, 2020, and 2021."

RC1: Line 212 : "...and marine" , really marine aerosol here? it does not make sense to even mention this. Probably you are referring to the optically based aerosols models that can be

distinguished with Aeronet. But the way this is phrased, it sounds like these aerosols are present.

AC: We thank the reviewer for highlighting this issue. The authors agree that this is confusing as the KLRS is far from the ocean. The Verma et al., (2015) threshold used here was for broad characterisation of the aerosols at the site and has been used in other scholarly articles for helping define thresholds (e.g., Bibi et al., 2016; Djossou et al., 2018; Iftikhar et al., 2018; Léon et al., 2021; Platero et al., 2018; Singh et al., 2020) . As HLMA is understudied, particularly using AERONET data, little to no thresholds had previously been defined. Therefore, the threshold of Verma et al., (2015) were used to broadly characterise what was happening at the site. Under this classification marine aerosols are present at the site, these maybe be due to clouds or other arid background aerosols at the site. Therefore, the authors have removed this mention of "marine aerosols" and added in a comment regard the use of this threshold later in the manuscript regarding this issue. The manuscript now reads:

"In this study, the likely presence of dust events was determined through use of initial generic thresholds at two different AERONET wavelengths, 500 nm and 1020 nm. Thresholds at 500 nm were used to broadly characterise aerosols at Lhù'ààn Mân with thresholds used in arid environments (Verma et al, 2015). Therefore, whilst the 500 nm definitions are useful to understand the aerosol environment at KLRS, it may not be truly representative of dust emissions. This is evident when compared to direct ground data observations, AERONET-derived dust events in this study recorded at longer wavelengths were found to be a closer match to the known frequency of events than those at shorter wavelengths (Figure 9ab).  For example, on a day where RC data shows dust events for 95.8% of the day (24th May 2018), 11.6% of AOD readings were classified as dust using the thresholds from Verma et al. (2015), whereas thresholds from Dubovik et al. (2002) yielded 24.2% AOD readings as dust. We, therefore, note that careful consideration in wavelength and definition thresholds is needed when quantifying HLMA in AERONET data." [Lines 500-511]

We hope this is satisfactory.

**Ref.:** Bibi, H., Alam, K., and Bibi, S.: In-depth discrimination of aerosol types using multiple clustering techniques over four locations in Indo-Gangetic plains, Atmos. Res., 181, 106–114, https://doi.org/10.1016/j.atmosres.2016.06.017, 2016.

Djossou, J., Léon, J. F., Barthélemy Akpo, A., Liousse, C., Yoboué, V., Bedou, M., Bodjrenou, M., Chiron, C., Galy-Lacaux, C., Gardrat, E., Abbey, M., Keita, S., Bahino, J., N'Datchoh, E. T., Ossohou, M., and Awanou, C. N.: Mass concentration, optical depth and carbon composition of particulate matter in the major southern West African cities of Cotonou (Benin) and Abidjan (Côte d'Ivoire), Atmos. Chem. Phys., 18, 6275–6291, https://doi.org/10.5194/acp-18-6275-2018, 2018.

Iftikhar, M., Alam, K., Sorooshian, A., Syed, W. A., Bibi, S., and Bibi, H.: Contrasting aerosol optical and radiative properties between dust and urban haze episodes in megacities of Pakistan, Atmos. Environ., 173, 157–172, https://doi.org/10.1016/j.atmosenv.2017.11.011, 2018.

Léon, J. F., Barthélémy Akpo, A., Bedou, M., Djossou, J., Bodjrenou, M., Yoboué, V., and Liousse, C.:

PM2.5surface concentrations in southern West African urban areas based on sun photometer and satellite observations, Atmos. Chem. Phys., 21, 1815–1834, https://doi.org/10.5194/acp-21-1815-2021, 2021.

Platero, I. Y., Estevan, R., Moya, A., and Yuli, R. A.: Determining the desert dust aerosol presence in the Mantaro Valley, Peru, Opt. Pura y Apl., 51, 1–14, https://doi.org/10.7149/OPA.51.3.50023, 2018.

Singh, P., Vaishya, A., Rastogi, S., and Babu, S. S.: Seasonal heterogeneity in aerosol optical properties over the subtropical humid region of northern India, J. Atmos. Solar-Terrestrial Phys., 201, 105246, https://doi.org/10.1016/j.jastp.2020.105246, 2020.

RC1: Figure 4: can you place location of video cameras in this figure?

AC: We thank the reviewer for this suggestion. The authors agree that this would aid with the readability of the figure. The locations of the video cameras were added to figure 3 &4 in the manuscript.

[Figure]

Figure 4. Image shows the plume rising from the Á'áy Chù delta and going out across the lake captured by PlanetScope (24th May 2018 at 11:58 am local time) overlaid point dust source locations (PDS). Due to the high resolution of the PlanetScope imagery (c. 3 m) we were able to trace the plume to the up-valley source on the delta. The Kluane Lake Research Station's position is identified in red (KLRS), with the RC side camera identified in purple and the island camera in green.

RC1: Table 2 is not referenced anywhere in the text. With respect MAIAC data, you could add the collection or version of the MAIAC algorithm.

AC: Thank you. Table 2 in now referenced throughout the text. The authors agree that the MAIAC version should be included, and it has been added to the manuscript.

**Table 2. Spectral bands and data quality of spectral data used in this study**

| Spectral Data Used | Wavelength | AERONET Data Quality Level | Application |
|---|---|---|---|
| AERONET AOD (AOD$_D$) | 1020 nm | 1 | Determination of DEDs |
| AERONET AOD (AOD$_A$) | 500 nm | 1 | Comparison against other aerosol types in air column |
| | | | Comparison against MODIS MAIAC |
| AERONET Angström exponent ($\alpha$) | 440-870 nm | 1 | Determination of DEDs |
| AERONET SSA | 440, 675, 870, and 1020 nm | 2 | Radiation scattering effectiveness of aerosols |
| AERONET Volume Size Distribution | 340, 380, 440, 500, 675, 870, 1020, and 1640 nm | 2 | The percentage of spherical particles in the observed aerosol to determine peaks in particle size |
| MODIS MAIAC (MCD19A2 V6.1) Land Aerosol Optical Depth Daily 1km (AOD$_M$) | 470 nm | n/a | Space-based AOD estimates |

RC1: 4.1 Event scale Observations. Can you please provide rough numerical estimate of the tops of the dust plumes? are we talking about tens of meters? a few hundred meters height? this is useful for contextual information.

AC: Thank you. Although not data analysed here, a lidar was installed in May 2019 at KLRS by co-authors (and the LiDAR is co-located with the AERONET station that we use here) and regularly sees plumes at or exceeding 500 m and in larger events exceeding 1 km.

RC1: Figure 6. This is a nice and informative figure. But what is the purpose of the labels a,b,c and d if they are not referenced in the text?

AC: We thank the reviewer for this suggestion. The authors agree with this comment and the labels are now referenced carefully throughout the text.

RC1: Also, please make clear in the x-axis that it is local time.

AC: We thank the reviewer for this suggestion. The authors agree that to make this figure more accessible an addition of local time was necessary. A revised version has been added to the manuscript.

[Figure]

RC1: Perhaps you could add in one of the mountain slopes a reference height to compare with the dust cloud? Also, the distance from cameras to mountain visible across the valley would be useful information.

AC: We thank the reviewer for this suggestion. The distance to the mountain from the side camera is ~5 km and the elevation of the small peak at the top-centre of the image frame is 1910 m asl or roughly 1100 m above the lake surface. As for the island camera, this is more challenging as the background is not as close. The closest set of mountains on the right of the frame that are snow-capped are 14 km away and 2270 m asl. Information covering these points has now been added to the manuscript in the figure 6 caption. The manuscript now reads:

"Figure 6. $AOD_D$ returns from the 24/05/2018 visualised with the corresponding oblique camera images during peak events. In the morning dust is in the southern section of delta. The distance to the mountain from the side camera (panels A&C) is ~5 km and the elevation of the small peak at the top-centre of the image frame is roughly 1100 m above the lake surface. As for the island camera (panels B&D), the closest set of mountains on the right of the frame that are snow-capped are 14 km away and 2270 m asl."

RC1: Line 359 - I found this reasoning difficult to follow because I could not see well in the images the camera locations.

AC: We thank the reviewer for this suggestion. The authors have updated the figures 3, 4, and 6 in order to clarify and better explain the camera locations in context with the dust source. See above for figures 4 and 6, figure 3 is updated below.

[Figure]

Figure 3. Locations of the oblique cameras and their approximate fields of view (3C). The Island camera in green is located on the former island in the delta looking south-west (3A). The Side camera is located near the Alaskan Highway looking north-west. Images so dust free views of the cameras (3B). Base imagery is from PlanetScope imagery in June 2018.

RC1: Line 400-404 I think it should be mentioned here the number of clear/cloudy days that Aeronet observed the Sun and how many of those dust was observed.

AC: We thank the reviewer for this suggestion. Due to the high latitude and mountainous area this site is situated in, it inevitably experiences a high amount of coincidence cloudy days. It would be hard to verify specific cloud impacts to a high-level of certainty, however, the authors have gone through MODIS Terra imagery and have added information on cloudy days for this daily overpass period to the supplementary information (Table S1).

**Table S1. Dust event days and cloudy days at Lhù'ààn Mân over the study period. Cloudy days were decerned by analysing MODIS Terra images and DED decerned using 1020 nm wavelength from Dubovik et al. (2002).**

|  | No. of cloudy days in month | No. of DEDs | No. of coincident DEDs and cloudy days |
|---|---|---|---|
| **May-18** | 24 | 17 | 14 |
| **Jun-18** | 23 | 26 | 20 |
| **Jul-18** | 21 | 17 | 13 |
| **Aug-18** | 24 | 13 | 17 |
| **Sep-18** | 17 | 13 | 9 |
| **Oct-18** | 24 | 10 | 10 |
| **May-19** | 24 | 19 | 15 |
| **Jun-19** | 28 | 20 | 19 |
| **Jul-19** | 31 | 17 | 17 |
| **Aug-19** | 28 | 20 | 19 |
| **Sep-19** | 16 | 15 | 9 |
| **Oct-19** | 5 | 1 | 0 |

RC1: Figure 8 Caption. The description is a bit difficult to read. Are the vertical bars the DED/week? Also, the coloured lines have poor contrast. Please consider changing and add the colour information in the caption too.

AC: We thank the reviewer for this suggestion. Agreed. The authors have updated the caption to aide this comment. The authors have also changed the colours to orange and blue for better contrast.

[Figure]

"Figure 8. Variability in DEDs in 2018 and selected seasonal variables that affect dust emission. The vertical bars display the total number of dust event days recorded by AERONET from $AOD_D$ per week. The AERONET station was recording from 14/05/18 until 21/10/18, but no data was recorded between 24/07/18 - 14/08/18. Average weekly snow depth (cm) displayed with the solid black line and average weekly air temperature (°C) displayed with the orange dot-dash line recorded at the Lhù'ààn Mân Research Site. Lake height displayed with the blue long dash line is taken from the Environment Canada lake depth gauge 09CA001 at Kluane Lake near Burwash Landing and is the average weekly water depth at that site."

RC1: Line 455-459. Please note that while relaxing the threshold criteria makes sense, it also introduces the possibility of cirrus contamination in the Aeronet data. I think and only in this case, it can be circumvented by inspecting the remote camera images for the days with Aeronet observations and check if there are cirrus in the background sky. This could be a quick and dirty way to check that Aeronet data is not contaminated.

AC: The authors thank the reviewer for comments on cirrus contamination. An investigation of available RC images to note the occurrence of cirrus cloud is currently being conducted by co-authors. At the time of the study, only one day (24/05/2018) of RC data was available to be analysed. To help mitigate the effects of cloud, the authors have investigated MODIS Terra imagery and added information on cloudy data to the supplementary information (Table S1).

RC1: Line 457. This is the first instance that Figure 9 is mentioned and it is referred in way as the reader is already familiar with the figure, which is not the case. So please rearrange the text to first introduce the figure and the refer to different sections of it.

AC: We thank the reviewer for this suggestion. The authors agree and the ordering of the manuscript was adjusted so that figure 9 comes earlier.

RC1: Lines 476-480 and 486-490. While I think it makes sense to use thresholds used in other Aeroent dust sites for this case, it is not entirely surprising that there are detection differences. First of all , this site is extremely close to the dust source something that not necessarily is the case in the reference sites used in lower latitudes. In particular, the rapid variability of dust concentrations in puffs of dust is probably one of the main differences. So for example, given the distance to the source, it is likely that this dust has a higher coarse mode contribution to the observed AOD and AE than in lower latitude sites. While I do not think that you can do much to improve on this, I do think that this fact should be mentioned and discussed as probable impacts in observed AODs and AEs.

AC: The authors agree with this statement and thank the reviewer for raising the issue. We have updated the discussion to mention and discuss these topics. The manuscript now reads:

"Further exploration of other AERONET products, for example the Spectral Deconvolution Algorithm (SDA), may further help define thresholds for HLMA. Furthermore, it is also important to note the location of the AERONET station relative to the dust source. KLRS is extremely close to the dust course which contributes to rapid variability in dust concentrations which will not be seen at mid-latitude locations far from source. It is likely that dust from this site will consequently have a higher coarse mode fraction that the AOD and AE at mid-latitude sites." [Lines 520-523]

RC1: Line 503-504. Not clear what you mean with "aerosol phases" , what are you referring to?

AC: We thank the reviewer for this suggestion. This was also raised by reviewer 2 and we have changed the text to 'aerosol types". The manuscript now reads:

"SSA was derived for the aerosol types show that dust scatter the most incoming radiation with biomass burning aerosols scattering slightly less (Figure 11c)."

RC1: Figure 11 Caption: Add a clarification that Aeronet retrievals of size distribution and SSA are carried out only for AOD>0.4.

AC: The authors agree, and the manuscript was updated to clarify this point. The manuscript now reads:

"Models used to calculate volume size distribution are based on AERONET level 2.0 DQ with and AOD > 0.4."

RC1: Figure 13: the way this is plotted, it suggests MODIS observed the area continuously which probably it did not happen. Can you add symbols to the days where there was a MODIS observation?

AC: We thank the reviewer for this suggestion. The authors have updated this figure to remove the continuous line of MAIAC data and have updated the figure with colours. Figure 13 has also been moved and can now be found in supplementary information figure S5.

[Figure]

RC1: General comment triggered by Figure 13

One reason why MODIS may have trouble in this place is that the MODIS pixels are too big, or the observed pixel contains variable combination of bright and dark surfaces all in one pixel that can't be accounted for the retrieval. So perhaps you could clarify somewhere the width of

the valley. For example, MODIS pixels are in the 500-1000m size. How do these compare with the typical size of the dust sources in the flood plain?

AC: Thank you. We can clarify this point as follows: The valley floor is 4-5 km wide. The MAIAC pixel size used in this study were 1km by 1km. The dust plume size is often bigger than this (around ~25 km$^2$). The manuscript now reads:

" 1 km by 1 km pixels within the 70km$^2$ southern portion of Lhù'ààn Mân was analysed in this study. The mean MAIAC AOD retrieval for each day was then used for analysis. The valley is 4-5 km wide, and the dust plume size is often bigger than this (around ~25 km$^2$). With dust advecting over the lake a uniform brightness background should aid in MAIAC AOD retrieval." [Lines 315- 319]

RC1: Perhaps, it would be illustrative to add a MODIS/VIIRS RGB of one event to illustrate how poorly the plumes are resolved (it will lucky very fuzzy). Just suggestion, it maybe informative for a presentation but probably take too much space in the manuscript.

AC: We thank the reviewer for this suggestion. The authors agree and have added an image of a MODIS captured dust plume into the supplementary information (Figure S4).

[Figure]

Figure S4: MODIS Terra Image of a dust plume blowing over KLRS on a clear day. Plume length (from end of delta to end of plume) is ~ 8km.

RC1: Line 579 ... detected by?

AC: The authors agree with this and the manuscript now reads:

"At Lhù'ààn Mân, an exceptional frequent emitter of dust, more than 97.8% of events are not being detected due to AERNET cloud screening algorithms."

---

## Author Comment (AC2)

**Author's response to reviewer number 2**

AC: We thank the anonymous referee for the detailed review of our manuscript. We carefully reviewed each comment and have amended at manuscript to address the issues raised. Reviewer comments are in black with our responses in blue. Changes to the manuscript are in the small font size 10.

RC2: General major comment:

- Besides the AOD data and the Angstrom Exponent data, did you analyze the SDA from the sun photometer? I believe this will give further information on fine and coarse mode AOD and it would help to better differentiate the fine and coarse mode aerosols.

AC: We thank the reviewer for this suggestion. We were aware that the SDA product can yield submicron (fine) and super-micron (coarse) AOD at 500 nm, and that from this the fraction of fine mode to total AOD can be computed. However, at the time we undertook this study, there were (and remain) few published dust detection case studies which had incorporated SDA data (e.g., Capelle et al., 2018; O'Niell et al., 2023). SDA products can also (like with regular AOD) produce significant errors when there is cirrus cloud, due to the high latitude and mountainous area this site is situated in, it inevitably experiences a high amount of incidence cloudy days. This is an issue with any AERONET data but it was decided that we would not use SDA. At the same time, we were also keen to use an approach that integrated short and long AERONET wavelengths in a comparable manner. To balance the discussion, we have included mention of the potential of SDA data going forward. We hope that this is satisfactory. The manuscript now reads:

"However, these thresholds may not encompass the optical parameters of HLMA and this may also impact retrievals at this site. Further exploration of other AERONET products, for example the Spectral Deconvolution Algorithm (SDA), may further help define thresholds for HLMA." [Lines 515-518]

**Ref**: Capelle, V., Chédin, A., Pondrom, M., Crevoisier, C., Armante, R., Crepeau, L., and Scott, N. A.: Infrared dust aerosol optical depth retrieved daily from IASI and comparison with AERONET over the period 2007–2016, Remote Sens. Environ., 206, 15–32, https://doi.org/10.1016/j.rse.2017.12.008, 2018.
O'Neill, N. T., Ranjbar, K., Ivanescu, L., Eck, T. F., Reid, J. S., Giles, D. M., Pérez-Ramírez, D., and Chaubey, J. P.: Relationship between the sub-micron fraction (SMF) and fine-mode fraction (FMF) in the context of AERONET retrievals, Atmos. Meas. Tech., 16, 1103–1120, https://doi.org/10.5194/amt-16-1103-2023, 2023.

- I think this classification based on the threshold from Verma et al. (2015) and Dubovik et al. (2002) are very primitive and as presented in figure 9b it is not very realistic. You try to classify aerosol types based on Verma et al. (2015) and Dubovik et al. (2002) thresholds. First of all, Dubovik et al. (2002) tried to investigate the absorption and other aerosol optical properties in several key locations. As they mentioned in this paper, they used these thresholds because the values of real and imaginary parts of the refractive index, as well as single scattering albedo, are given only for the condition of $\tau_{(440)} > 0.4$ for Urban-industrial, mixed, and biomass burning aerosols, and for the conditions of $\tau_{ext(1020)} > 0.3$ and $\alpha < 0.6$ for desert dust. On the other hand, Verma et al. (2015) tried to define thresholds for the Jaipur AERONET station in India and the aerosol in different regions has its own properties (as you

claimed in the introduction). So, I strongly suggest authors define their thresholds for their station based on the properties of the aerosol at Kluane lake station.

AC: We thank the reviewer for this comment. The Verma et al. (2015) threshold used in this study was for broad characterisation of the aerosols at the site and has been used in other scholarly articles for helping define thresholds (e.g., Bibi et al., 2016; Djossou et al., 2018; Iftikhar et al., 2018; Léon et al., 2021; Platero et al., 2018; Singh et al., 2020). The Dubovik et al. (2002) paper is a highly cited paper used for defining thresholds for dust using AERONET which verified AERONET data using in-situ measurements (e.g., Ciren and Kondragunta, 2014). As HLMA is understudied, particularly using AERONET data, little to no thresholds had previously been defined. Therefore, the thresholds of Verma et al. (2015) were used to broadly characterise what was happening at the site. Under this classification marine aerosols are present at the site, these maybe be due to clouds or other arid background aerosols at the site. Therefore, the authors added in a comment regard the use of this threshold later in the manuscript regarding this issue. The manuscript now reads:

"The Verma et al. (2015) threshold is used in this study for broad characterisation of the aerosols at the site and has been used in other scholarly articles for helping define thresholds (e.g., Bibi et al., 2016; Djossou et al., 2018; Iftikhar et al., 2018; Léon et al., 2021; Platero et al., 2018; Singh et al., 2020). Under the Verma et al. (2015) classification, marine aerosols are flagged, but at KLRS we concur and assume that these possibly represent clouds and/or other arid background aerosols present at the site (see Table 1)." [Lines 250-255]

"In this study, the likely presence of dust events was determined through use of initial generic thresholds at two different AERONET wavelengths, 500 nm and 1020 nm. Thresholds at 500 nm were used to broadly characterise aerosols at Lhù'ààn Mân with thresholds used in arid environments (Verma et al, 2015). Therefore, whilst the 500 nm definitions are useful to understand the aerosol environment at KLRS, it may not be truly representative of dust emissions. This is evident when compared to direct ground data observations, AERONET-derived dust events in this study recorded at longer wavelengths were found to be a closer match to the known frequency of events than those at shorter wavelengths (Figure 9ab). For example, on a day where RC data shows dust events for 95.8% of the day (24th May 2018), 11.6% of AOD readings were classified as dust using the thresholds from Verma et al. (2015), whereas thresholds from Dubovik et al. (2002) yielded 24.2% AOD readings as dust. We, therefore, note that careful consideration in wavelength and definition thresholds is needed when quantifying HLMA in AERONET data." [Lines 500 -511]

We hope this is satisfactory.

Ref.: Arola, A., Eck, T. F., Kokkola, H., Pitkänen, M. R. A., and Romakkaniemi, S.: Assessment of cloud-related fine-mode AOD enhancements based on AERONET SDA product, Atmos. Chem. Phys., 17, 5991–6001, https://doi.org/10.5194/acp-17-5991-2017, 2017.

Bibi, H., Alam, K., and Bibi, S.: In-depth discrimination of aerosol types using multiple clustering techniques over four locations in Indo-Gangetic plains, Atmos. Res., 181, 106–114, https://doi.org/10.1016/j.atmosres.2016.06.017, 2016.

Ciren, P. and Kondragunta, S.: Journal Dust aerosol index (DAI) algorithm for MODIS, J. Geophys. Res., 119, 6196–6206, https://doi.org/10.1002/2014JD021606, 2014.

Djossou, J., Léon, J. F., Barthélemy Akpo, A., Liousse, C., Yoboué, V., Bedou, M., Bodjrenou, M., Chiron, C., Galy-Lacaux, C., Gardrat, E., Abbey, M., Keita, S., Bahino, J., N'Datchoh, E. T., Ossohou, M., and Awanou, C. N.: Mass concentration, optical depth and carbon composition of particulate matter in the major southern West African cities of Cotonou (Benin) and Abidjan (Côte d'Ivoire), Atmos. Chem. Phys., 18, 6275–6291, https://doi.org/10.5194/acp-18-6275-2018, 2018.

Iftikhar, M., Alam, K., Sorooshian, A., Syed, W. A., Bibi, S., and Bibi, H.: Contrasting aerosol optical and radiative properties between dust and urban haze episodes in megacities of Pakistan, Atmos. Environ., 173, 157–172, https://doi.org/10.1016/j.atmosenv.2017.11.011, 2018.

Léon, J. F., Barthélémy Akpo, A., Bedou, M., Djossou, J., Bodjrenou, M., Yoboué, V., and Liousse, C.: PM2.5surface concentrations in southern West African urban areas based on sun photometer and satellite observations, Atmos. Chem. Phys., 21, 1815–1834, https://doi.org/10.5194/acp-21-1815-2021, 2021.

Platero, I. Y., Estevan, R., Moya, A., and Yuli, R. A.: Determining the desert dust aerosol presence in the Mantaro Valley, Peru, Opt. Pura y Apl., 51, 1–14, https://doi.org/10.7149/OPA.51.3.50023, 2018.

Singh, P., Vaishya, A., Rastogi, S., and Babu, S. S.: Seasonal heterogeneity in aerosol optical properties over the subtropical humid region of northern India, J. Atmos. Solar-Terrestrial Phys., 201, 105246, https://doi.org/10.1016/j.jastp.2020.105246, 2020.

- If you are using Version 3 Level 1.0 data (or DQ 1) you need to further investigate the clouds effects on your results. I quickly checked MODIS Aqua imageries for Kluane lake for the month of May 2018 and based on these images at least 30% of the days in this month were cloudy or partly cloudy.

AC: We thank the reviewer for this suggestion. Due to the high latitude and mountainous area this site is situated in, it inevitably experiences a high amount of incidence cloudy days. It would be hard to verify specific cloud impacts to a high-level of certainty. An investigation of available RC images to note the occurrence of cirrus cloud is currently being investigated by co-authors. At the time of the study, only one day (24/05/2018) of RC data was available to be analysed. However, the authors have gone through MODIS Terra imagery and have added information on cloudy days for this daily overpass period to the supplementary information (Table S1). The supplementary information now reads:

**"S3. Implications of using Level 1.0 AERONET data**

Cloud-screening is an essential part of the AERONET network data refinement. Clouds affect AOD, with cirrus clouds being included in the fine mode AOD and other cloud types having larger optical depths (Arola et al., 2017). To extract errors where the clouds are thin enough to get direct sun measurements, cloud screening is applied to all AEROENT data (Arola et al., 2017). This cloud screening removes around ~20 to 50% of the data (Smirnov et al., 2000), and as seen in figure 9a removes many dust events. To get an accurate image of dust events at the site, level 1.0 data is used, but as a by-product some cloud optical depth may also be included in AOD

measurements. Arola *et al*. (2017) found that the uncloud-screened data added roughly 0.007 and 0.0012 onto the AOD. However, these results are proprietary and due to being unable to calculate the effect that cloud screening has at Lhù'ààn Mân and other locations or whether the screened cloud was actually dust, uncloud-screened data was used throughout the study. The SSA and volume size distribution inversion products are run using level 2.0 cloud-screened data. Days recorded as DEDs may have had some of the dust aerosol scans removed by these products and subsequent results.

**Table S1. Dust event days and cloudy days at Lhù'ààn Mân over the study period. Cloudy days were decerned by analysing MODIS Terra images and DED decerned using 1020 nm wavelength from Dubovik et al. (2002).**

|  | No. of cloudy days in month | No. of DEDs | No. of coincident DEDs and cloudy days |
|---|---|---|---|
| **May-18** | 24 | 17 | 14 |
| **Jun-18** | 23 | 26 | 20 |
| **Jul-18** | 21 | 17 | 13 |
| **Aug-18** | 24 | 13 | 17 |
| **Sep-18** | 17 | 13 | 9 |
| **Oct-18** | 24 | 10 | 10 |
| **May-19** | 24 | 19 | 15 |
| **Jun-19** | 28 | 20 | 19 |
| **Jul-19** | 31 | 17 | 17 |
| **Aug-19** | 28 | 20 | 19 |
| **Sep-19** | 16 | 15 | 9 |
| **Oct-19** | 5 | 1 | 0 |

**Ref.:** Arola, A., Eck, T. F., Kokkola, H., Pitkänen, M. R. A., and Romakkaniemi, S.: Assessment of cloud-related fine-mode AOD enhancements based on AERONET SDA product, Atmos. Chem. Phys., 17, 5991–6001, https://doi.org/10.5194/acp-17-5991-2017, 2017.

- I suggest adding a few sentences about how you compare daily mean MAIAC AOD with daily AERONET mean AOD. It is not clear if you compare one pixel (1*1km) with AERONET AOD or if you chose a bigger area and then averaged AOD values and used the averaged value for your analysis.

AC: We thank the reviewer for this suggestion. We created a polygon over the southern portion of Lhù'ààn Mân and calculated the average. This has been updated in the manuscript and now reads:

"1 km by 1 km pixels within the $70km^2$ southern portion of Lhù'ààn Mân was analysed in this study. The mean MAIAC AOD retrieval for each day was then used for analysis. The valley is 4-5 km wide, and the dust plume size is often bigger than this (around ~25 $km^2$ – see figure S4). With dust advecting over the lake a uniform brightness background should aid in MAIAC AOD retrieval." [Lines 315- 319]

We hope this is satisfactory.

RC2: Specific comments:

RC2: I suggest using either "high latitude" or "high-latitude"

AC: We thank the reviewer for this suggestion. The authors agree and have amended all cases to high-latitude.

RC2: I suggest using either "ground based" or "ground-based"

AC: We thank the reviewer for this suggestion. The authors agree and have amended all cases to ground-based.

RC2: L94: by direct sun and "sky scan" measurements

AC: We thank the reviewer for this suggestion. This was a useful comment and the authors have updated the manuscript accordingly. The manuscript now reads:

"The Aerosol Robotic Network (AERONET) is a ground-based collaborative network of automated sun-sky scanning spectral radiometers that determine the aerosol optical and microphysical properties by direct sun and "sky-scan" measurements (Holben, 1998)".

RC2: L98: The Angström exponent ($\alpha$) allows estimation of aerosol particle size (effective radius) not aerosol size distribution.

AC: We thank the reviewer for this suggestion. This has been amended in the manuscript and noted by the authors. The manuscript now reads:

"The spectral aerosol optical depth determined from these data are also used to derive an Angström exponent ($\alpha$) which can in turn allow estimation of aerosol particle size (O'Neill et al., 2003)".

RC2: L103: this map needs to be fixed. I don't see any highlighted area in "yellow" and also circles are outside of the map. In the legend N is ~1750 but, in the caption, it says N ~ 1075. Please fix this as well.

RC: The authors thank the reviewer for highlighting the inconsistencies in this figure. The figure has now been modified to a more simplified format by changing the colours and caption, updating the numbers, and correcting the projection.

[Figure]

Figure 1. Location of all AERONET stations with stations in the high-latitudes or proglacial areas highlighted in orange. Cryospheric stations account for 48 out of 1655 global AERONET (data from the AREONET website: https://aeronet.gsfc.nasa.gov) .

RC2: L134-135 Since AERONET also is a remote sensing measurement maybe you need to specify the remote observations to satellite observations.

AC: We thank the reviewer for this suggestion. The authors carefully considered this comment and concluded that we differentiate between remote observations and satellite observations by referring to "space-based" and "ground-based" measurements.

RC2: L199: Please add "Figure S1"

AC: We thank the reviewer for this suggestion. The authors agree and added figure S1 into the manuscript. The manuscript now reads:

"This is evidenced in meteorological stations further up and down the valley which are much less directionally variable than KLRS, with dominant wind directions of North-Northeast (supplementary information figure S1)."

RC2: L 210: You need to specify these max and min observed values are for the Kluane Lake AERONET station.

AC: We thank the reviewer for this suggestion. The authors agree and have added max and min to the AERONET stations. The manuscript now reads:

 "AERONET returns at 1.0 data quality (DQ) range for between 232 measurements per day (maximum observed on 21/06/2019) to 1 return per day (minimum observed on 06/07/2018) at KLRS."

RC2: L212: One of the aerosol types that you have in your classification is Marine. I don't know if it is the case for Kluane lake with about 200 km far from the ocean and all of those mountains around the lake.

AC: We thank the reviewer for highlighting this issue. The authors agree that this is confusing as the KLRS is far from the ocean. The Verma et al., (2015) threshold used here was developed for a different location but used here to provide a broad characterisation of the detectable aerosols at the site. Under the Verma classification, marine aerosols were initially flagged, but at KLRS we concur and assume that these *actually* represent clouds and/or other arid background aerosols present at the site. Therefore, the authors have removed mention of "marine aerosols" and added a comment to underpin this change to the Verma approach later in the manuscript. The manuscript now reads:

"In this study, the likely presence of dust events was determined through use of initial generic thresholds at two different AERONET wavelengths, 500 nm and 1020 nm. Thresholds at 500 nm were used to broadly characterise aerosols at Lhù'ààn Mân with thresholds used in arid environments (Verma et al, 2015). Therefore, whilst the 500 nm definitions are useful to understand the aerosol environment at KLRS, it may not be truly representative of dust emissions. This is evident when compared to direct ground data observations, AERONET-derived dust events in this study recorded at longer wavelengths were found to be a closer match to the known frequency of events than those at shorter wavelengths (Figure 9ab). For example, on a day where RC data shows dust events for 95.8% of the day (24th May 2018), 11.6% of AOD readings were classified as dust using the thresholds from Verma et al. (2015), whereas thresholds from Dubovik et al. (2002) yielded 24.2% AOD readings as dust. We, therefore, note that careful consideration in wavelength and definition thresholds is needed when quantifying HLMA in AERONET data." [Lines 500-511]

We hope this is satisfactory.

RC2: L 213-214: based on Giles et. al., (2019), Level 1.5 represents near-real-time automatic cloud screening and automatic instrument anomaly quality controls and Level 2.0 additionally applies pre-field and post-field calibrations.

AC: We thank the reviewer for this suggestion and have now noted this detail in the text. The manuscript now reads:

"AERONET AOD data are computed at three DQ levels: Level 1.0 (unscreened), level 1.5 (represents near-real-time automatic cloud screening and automatic instrument anomaly quality controls), and Level 2.0 (all of above and applies pre-field and post-field calibrations) with the Version 3 automated control algorithm."

RC2: L222-225: So how about cases where AOD is less than 0.3? That is the reason I think you should use the SDA product which is a standard product of AERONET and it breakdown the total AOD into fine and coarse mode AOD. So, you don't need to define this threshold.

AC: We thank the reviewer for this suggestion. Due to the close proximity of the AERONET station to the dust source, having AOD readings of lower than 0.3 is not an issue in this context. We were aware that the SDA product can yield submicron (fine) and super-micron (coarse) AOD at 500 nm, and that from this the fraction of fine mode to total AOD can be computed. However, at the time we undertook this study, there were (and remain) few published dust

detection case studies which had incorporated SDA data. At the same time, we were also keen to use an approach that integrated short and long AERONET wavelengths in a comparable manner. To balance the discussion, we have included mention of the potential of SDA data going forward. We hope that this is satisfactory. The manuscript now reads:

"However, these thresholds may not encompass the optical parameters of HLMA and this may also impact retrievals at this site. Further exploration of other AERONET products, for example the SDA, may further help define thresholds for HLMA." [Lines 515-518]

RC2: L310-312: Please add an explanation of why you use Spearman's rank correlation coefficient.

AC: We thank the reviewer for noting the need for clarification. A Spearman's rank correlation was used here to underpin likely relationships between variables and the approach conforms to assumptions regarding the data set and the data distribution. However, the authors agree that this does not aid the narrative of the paper, and we have therefore removed it.

RC2: L397: Not only below the mountain line but also if dust remains out of the sun photometer's FOV still it will not be captured. So, for example, if the sun is in the south of the KLRS and the dust plume is to the north of the site, still it will not be captured.

AC: Thank you for drawing this to the authors attention an explanation of this was added to the manuscript. The manuscript now reads:

"A possible explanation as to why AERONET detected dust in the afternoon when the plume was not originating in the southern section of the delta is that the vertical motion of emitted dust was greater so that it was above the mountain line and thus detectable by AERONET. Furthermore, AERONET requires a direct sun measurement, if it is cloudy the dust will not be recorded. Dust must be in the AERONET stations FOV, when the sun is south of KLRS and the dust plume advects north, it will not be captured. A combination of dust emission in the evening and low height of the dust lead to dust event's detection being underrepresented at Lhù'ààn Mân." [Lines 409-415]

RC2: L413-419: the caption is not very clear please use the same definition for the different variables as your y-axis labels. For example, you talk about Average weekly depth and the y labels are Lake Depth (m) and Weekly total Snowfall (cm) and it is a bit confusing.

AC: The authors agree, and this figure was fixed, the addition of weekly mean lake depth was added, and colours of lines changed for readability. The caption was also update accordingly and now reads:

[Figure]

Figure 8. Variability in DEDs in 2018 and selected seasonal variables that affect dust emission. The vertical bars display the total number of dust event days recorded by AERONET from $AOD_D$ per week. The AERONET station was recording from 14/05/18 until 21/10/18, but no data was recorded between 24/07/18 - 14/08/18. Average weekly depth (cm) displayed with the solid line and average weekly air temperature (°C) displayed with the dot-dash line recorded at the Lhù'ààn Mân Research Site. Lake height displayed with the long dash line is taken from the Environment Canada lake depth gauge 09CA001 at Kluane Lake near Burwash Landing and is the average weekly water depth at that site.

RC2: L428: In the Figure 8 caption you said that "The AERONET station was recording from 14/05/18 until 21/10/18" and here "The AERONET station began recording on 13th May 2018". Please fix the one that is not correct.

AC: We thank the reviewer for this suggestion. This was corrected to 14/05/18 until 21/10/18.

RC2: L480: I don't agree with this sentence. The reason that shorter wavelengths lead to an underrepresentation of dust events and DED frequency is the threshold that you used for the shorter wavelength.

AC: We thank the reviewer for this comment and the authors agree, the 500 nm threshold was used to broadly characterise and may not be reflective of actual HLMA at the site. We have now updated the manuscript and the whole paragraph reads:

"Given the distribution of monitoring sites, most previous studies which investigate the radiative signature of dust using AERONET returns have been conducted at low latitudes (desert dust) using DQ 1.5 AERONET data (e.g., Santese et al., 2013; Binietoglou et al., 2015). In this study, the likely presence of dust events was determined through use of initial generic thresholds at two different AERONET wavelengths, 500 nm and 1020 nm.

Thresholds at 500 nm were used to broadly characterise aerosols at Lhù'ààn Mân with thresholds used in arid environments (Verma et al, 2015). Therefore, whilst the 500 nm definitions are useful to understand the aerosol environment at KLRS, it may not be truly representative of dust emissions. This is evident when compared to direct ground data observations, AERONET-derived dust events in this study recorded at longer wavelengths were found to be a closer match to the known frequency of events than those at shorter wavelengths (Figure 9ab). For example, on a day where RC data shows dust events for 95.8% of the day (24th May 2018), 11.6% of AOD readings were classified as dust using the thresholds from Verma et al. (2015), whereas thresholds from Dubovik et al. (2002) yielded 24.2% AOD readings as dust. We, therefore, note that careful consideration in wavelength and definition thresholds is needed when quantifying HLMA in AERONET data." [Lines 500-511]

RC2: L493-494: Please rewrite this sentence. It is hard to understand.

AC: We thank the reviewer for this comment. The sentence has been rewritten to improve readability. The manuscript now reads:

"For DEDs in 2018 (using definitions from Dubovik *et al.* 2002 and at DQ 1.0) at Lhù'ààn Mân, the average α was -0.003 indicating a very coarse grain size."

RC2: L496-497: based on Figures 11a and 11b the distribution is not bimodal and it is trimodal with a fine mode peak of 0.2 to 0.4 μm and 2 coarse mode peaks at 2.6 and 10.1μm.

AC: We thank the reviewer for this suggestion. The authors agree with this and have amended the manuscript to trimodal accordingly. The manuscript now reads:

" Scans made on DEDs (Figure 11a) show a trimodal distribution with peaks at 2.6 and 10.1μm."

RC2: L499_500: AERONET inversion PSD bins (x-axis) are radius, not diameter but Bachelder et al. (2020) results as you mentioned here are in diameter so you need to convert one to another and then compare them together.

AC: The data from AERONET inversions are indeed in radius. This was then converted to diameter by the authors and always referred to as diameter to match Bachelder et al., (2020) and is common in scholarly work (e.g., Huang et al., 2023; Ryder et al., 2019; Song et al., 2022).

**Ref:** Huang, Y., Kok, J. F., Saito, M., and Muñoz, O.: Single-scattering properties of ellipsoidal dust aerosols constrained by measured dust shape distributions, Atmos. Chem. Phys., 23, 2557–2577, https://doi.org/10.5194/acp-23-2557-2023, 2023.
Ryder, C. L., Highwood, E. J., Walser, A., Seibert, P., Philipp, A., and Weinzierl, B.: Coarse and giant particles are ubiquitous in Saharan dust export regions and are radiatively significant over the Sahara, Atmos. Chem. Phys., 19, 15353–15376, https://doi.org/10.5194/acp-19-15353-2019, 2019.
Song, Q., Zhang, Z., Yu, H., Kok, J. F., Di Biagio, C., Albani, S., Zheng, J., and Ding, J.: Size-resolved dust direct radiative effect efficiency derived from satellite observations, Atmos. Chem. Phys., 22, 13115–13135, https://doi.org/10.5194/acp-22-13115-2022, 2022.

RC2: L501: Do you think this second peak (around 10μin radius) is real or it might be affected by the cloud? I know these results are from Level 2 inversion but if the cloud screening doesn't work well for direct sun measurements, how much do you believe in these results from AERONET Inversion?

AC: The authors thank the reviewer for this comment. We took into careful consideration this comment. The cloud screening algorithm does not work well for dust detection as classifies dust as cloud and consequently removes the dust, we do not envision that it works vice versa. Therefore, we can assume there is some course dust blowing from the site which may be responsible for this second peak. We hope that this is satisfactory.

RC2: L503: "aerosol phases" is not clear to me. I think by "aerosol phases" you mean "aerosol types"

AC: The authors thank the reviewer for this comment, which was also raised by reviewer 1. The manuscript has been updated to aerosol types. The manuscript now reads:

"SSA was derived for the aerosol types show that dust scatter the most incoming radiation with biomass burning aerosols scattering slightly less (figure 11c)."

L510: the colours in the scatterplot are not the same as the colour in the caption, please fix this.

AC: The authors thank the reviewer for raising this and the caption has been amended to reflect the colours in the scatterplot and now reads:

[Figure]

"Figure 9. (a)  The number of DEDs by month from 1020 nm over the 2018 and 2019 Lhù'ààn Mân dust seasons detected by the different data quality levels computed by AERONET. DQ 1.0 is represented in blue, DQ 1.5 is represented in orange, and DQ 2.0 in green. (b) Scatterplot of daily DQ 1.0 AOD$_A$ and α at 440-870nm used to broadly characterise aerosols at Lhù'ààn Mân for DEDs in the dust seasons of 2018 and 2019. In blue is the dust, black is biomass/industrial aerosols, grey is the arid background, green is marine type aerosols, and mixed types are represented in orange. Thresholds for each class are taken from Verma et al. (2015) whose definitions stems from Jaipur in India, these thresholds may vary from those found a KLRS but give a broad characterisation of aerosols present."

RC2: L519: Please fix the colour or the caption of Figure 10. I don't see any yellow or black in this plot.

AC: The authors thank the reviewer for raising this and the caption has been amended to reflect the colours in the scatterplot. The caption now reads:

"Figure 10. Aerosol compositions comparisons at Lhù'ààn Mân on a DED - 24/05/2018 (A&B) and a non-dust event day- 23/07/2018 (C&D). Scatterplots A&C are at DQ 1.0 AOD at 500 nm with thresholds from Verma et al. (2015) for broad characterisation of aerosols present at the site. The scatterplots B&D are at DQ 1.0 $AOD_D$ with thresholds from Dubovik et al. (2002). In blue is the dust, grey is the arid background, green is marine aerosols, and mixed types are represented in orange."

L521: Figure 11 a and b x-axis labels are radius, not diameter, please fix the labels.

AC: We thank the reviewer for this suggestion. Calculations were converted to diameter for these plots for easy comparison with other works.

RC2: L521: Why some of these plots in figure 11 are just for one year (for example Figure 11c) and some of these plots are for two years (Figures 11d and e)? Maybe it would be more reasonable if you keep the time range the same for all plots.

AC: We thank the reviewer for this comment. The inversion products were only available at KLRS from 2019 onwards. We have, therefore, amended the figure to only reflect plots from 2019 and updated the caption.

[Figure]

Figure 11. (a) Volume size distribution of aerosols at Lhù'ààn Mân Recordings that were taken on a DED (DED definition from Dubovik et al. (2002) in the 1020 nm band) are in blue, and recordings taken when there were biomass burning (defined from Verma et al., 2015) events (smoke) are in grey. Any days where dust or smoke

were not detected in AERONET are displayed in black. Models used to calculate volume size distribution are based on AERONET level 2.0 DQ with and AOD > 0.4. Dust event days are then pulled out for (b) underneath to show the presence of trimodal dust size. (c) Single scattering albedo for the principal aerosol types recorded at the Lhù'ààn Mân AERONET station for 2019 when AOD >0.4. Dust events are in green, biomass burning events are in pink, all other aerosol types are in blue. Models that calculate SSA are based on AERONET level 2.0 DQ. Frequency histograms of AOD (d) and Angstrom exponent (e) at Kluane Lake AERONET station in 2019.

RC2: L527: Again, the distribution is trimodal with a fine mode peak and two coarse mode peaks.

AC: We thank the reviewer for this suggestion. Thank you for this comment the manuscript was updated accordingly. The manuscript now reads:

"Dust event days are then pulled out for (b) underneath to show the presence of trimodal dust size."

RC2: L529-530: Again, the colours in the caption have not matched the colours in the plot.

AC: Thank you for this comment the manuscript was updated accordingly. The caption now reads:

"Figure 11. (a) Volume size distribution of aerosols at Lhù'ààn Mân Recordings that were taken on a DED (DED definition from Dubovik et al. (2002) in the 1020 nm band) are in blue, and recordings taken when there were biomass burning (defined from Verma et al., 2015) events (smoke) are in grey. Any days where dust or smoke were not detected in AERONET are displayed in black. Models used to calculate volume size distribution are based on AERONET level 2.0 DQ with and AOD > 0.4. Dust event days are then pulled out for (b) underneath to show the presence of trimodal dust size. (c) Single scattering albedo for the principal aerosol types recorded at the Lhù'ààn Mân AERONET station for 2019 when AOD >0.4. Dust events are in green, biomass burning events are in pink, all other aerosol types are in blue. Models that calculate SSA are based on AERONET level 2.0 DQ. Frequency histograms of AOD (d) and Angstrom exponent (e) at Kluane Lake AERONET station over 2018 and 2019."

RC2: L554: I think you mean Figures 12 and 13 .

AC: The authors thank the reviewer for this comment, the authors agree and updated the manuscript. The manuscript now reads:
"Figures 12 and 13 suggest a weak relationship between $AOD_A$ and $AOD_M$ at Lhù'ààn Mân during 2018."